# Distinct modes of SMAD2 chromatin binding and remodeling shape the transcriptional response to NODAL/ Activin signaling

Davide M Coda[1†], Tessa Gaarenstroom[1†‡], Philip East[2], Harshil Patel[2], Daniel S J Miller[1], Anna Lobley[2], Nik Matthews[3], Aengus Stewart[2], Caroline S Hill[1]*

[1]Developmental Signalling Laboratory, The Francis Crick Institute, London, United Kingdom; [2]Bioinformatics and Biostatistics, The Francis Crick Institute, London, United Kingdom; [3]Advanced Sequencing, The Francis Crick Institute, London, United Kingdom

*For correspondence: caroline.hill@crick.ac.uk

†These authors contributed equally to this work

Present address: ‡Wellcome Trust/Cancer Research UK Gurdon Institute, Cambridge, United Kingdom

Competing interests: The authors declare that no competing interests exist.

**Abstract** NODAL/Activin signaling orchestrates key processes during embryonic development via SMAD2. How SMAD2 activates programs of gene expression that are modulated over time however, is not known. Here we delineate the sequence of events that occur from SMAD2 binding to transcriptional activation, and the mechanisms underlying them. NODAL/Activin signaling induces dramatic chromatin landscape changes, and a dynamic transcriptional network regulated by SMAD2, acting via multiple mechanisms. Crucially we have discovered two modes of SMAD2 binding. SMAD2 can bind pre-acetylated nucleosome-depleted sites. However, it also binds to unacetylated, closed chromatin, independently of pioneer factors, where it induces nucleosome displacement and histone acetylation. For a subset of genes, this requires SMARCA4. We find that long term modulation of the transcriptional responses requires continued NODAL/Activin signaling. Thus SMAD2 binding does not linearly equate with transcriptional kinetics, and our data suggest that SMAD2 recruits multiple co-factors during sustained signaling to shape the downstream transcriptional program.

## Introduction

Patterning of tissues during embryonic development depends on the control of gene expression by extracellular signals. The signaling molecules involved frequently act as morphogens to impart positional information (*Ashe and Briscoe, 2006*). It is becoming clear that the resultant programs of gene expression evolve over time, and that transcriptional output is less a function of absolute ligand concentration, than duration of ligand exposure and signal transduction dynamics (*Cohen et al., 2013*; *Nahmad and Lander, 2011*). Genome-wide interrogation of transcription factor (TF) binding sites and distribution of histone modifications in many different cell types has greatly informed our knowledge of the mechanisms underlying transcription regulation (*van Dijk et al., 2014*). However, most of these studies have been performed at steady state and we know little about the sequence of events that occur from TF binding to the induction of dynamic programs of gene expression.

This is best studied using TFs that are activated in response to extracellular signaling and the NODAL signaling pathway provides an excellent model system. NODAL is a member of the transforming growth factor β (TGF-β) superfamily which plays crucial roles in early embryonic development. It is required at low levels for maintenance of pluripotency in both early embryos and

**eLife digest** To allow a complex animal to develop from a small bundle of cells, the cells need to be able to communicate with each other to coordinate their activities. Furthermore, this communication needs to continue in adulthood to keep the body in balance and to prevent diseases such as cancer.

The cells communicate by releasing signals that influence the behavior of their neighbors by activating proteins called transcription factors. These proteins then change the activity of particular genes in the nucleus by binding to specific places on a structure called chromatin (the structure in which the genes are packaged). One group of signaling molecules is known as the transforming growth factor beta superfamily, which is crucial for embryos to develop correctly. Failure to control these signals can also promote the growth of tumors. However, it is not clear how the detection of these signals at the surface of the cell leads to changes in the activity of genes inside the nucleus.

Two transforming growth factor beta signals called Activin and NODAL cause a transcription factor known as SMAD2 to move into the nucleus where it can alter gene activity. Here Coda, Gaarenstroom et al. investigated how SMAD2 transmits the Activin/Nodal signal in mouse cancer cells. The experiments showed that SMAD2 can change the activities of genes in multiple ways. SMAD2 can bind to places in the chromatin that are either easy to access (which typically contain genes that are already "switched on") as well as areas that are difficult to access (which generally contain genes that are "switched off"). As a result, SMAD2 increases the activity of genes that were already active, but also switches on on genes that were previously inactive.

Coda, Gaarenstroom et al. also found evidence that SMAD2 remained bound to chromatin after long periods of Activin/NODAL signaling. For some genes, this resulted in high gene activity, but in other cases this decreased the gene's activity. Therefore, future experiments will investigate which other proteins help SMAD2 to change gene activity at later times.

embryonic stem cells (ESCs), and at higher levels for mesendoderm differentiation and left–right patterning (*Arnold and Robertson, 2009*; *Schier, 2009*). Moreover, in zebrafish embryos, the time of ligand exposure dictates the cell fates specified, with endoderm requiring a longer duration of NODAL signaling than mesoderm (*Hagos and Dougan, 2007*). How one signal can induce both self-renewal and differentiation to different cell fates is not understood.

NODAL and the highly-related ligand Activin, which in vitro mimics the functional activities of NODAL, signal via heterotetrameric receptor complexes. These comprise type I and type II receptors, with NODAL additionally requiring the co-receptor TDGF1 (previously called Cripto) (*Schier, 2009*). Ligand binding leads the type II receptor ACVR2A/ACVR2B to phosphorylate and activate the type I receptor ACVR1B, which in turn recruits and phosphorylates the intracellular transducers, SMAD2 and SMAD3 (*Wu and Hill, 2009*). Once phosphorylated, SMAD2/3 form complexes with SMAD4 and accumulate in the nucleus, where they directly regulate transcription (*Ross and Hill, 2008*). SMAD3 and SMAD4 contact DNA through their N-terminal Mad homology 1 (MH1) domain, recognizing the sequence AGAC/GTCT (*Zawel et al., 1998*). SMAD2, in contrast, cannot bind DNA directly (*Yagi et al., 1999*) and requires interaction with additional TFs to recruit it to DNA (*Gaarenstroom and Hill, 2014*). The Forkhead TF FOXH1 was the first such cooperating TF to be identified, and is essential for the induction of a subset of NODAL/Activin target genes during gastrulation (*Attisano et al., 2001*; *Chen et al., 1996*). Other SMAD2-recruiting TFs include MIXER, POU5F1/OCT4, NANOG, EOMES and TEAD proteins, which have distinct DNA binding specificities compared with FOXH1 (*Beyer et al., 2013*; *Brown et al., 2011*; *Faial et al., 2015*; *Germain et al., 2000*; *Kunwar et al., 2003*; *Mullen et al., 2011*). It remains unclear how activated SMAD complexes find their binding sites on DNA, in particular, whether the recruiting TFs are pre-bound or whether they bind simultaneously with the SMADs. One study proposed that SMAD complexes can only bind to sites already occupied by Master TFs (*Mullen et al., 2011*). However, this model ascribes a purely modulatory role for TGF-$\beta$ superfamily pathways, which is at odds with the clear driving role of NODAL/Activin signaling in early development (*Schier, 2009*).

We previously established that SMAD2–SMAD4 complexes are unable to activate transcription on a naked DNA template, but instead require the context of chromatin (*Ross et al., 2006*), indicating that SMADs regulate gene expression through chromatin remodeling. Consistent with this idea the SMADs are known to recruit a number of chromatin modifiers and remodelers including the histone acetyltransferase (HAT) EP300 (previously called p300), the ATP-dependent helicase SMARCA4 (previously called BRG1), the H3K27 demethylase JMJD3 and Mediator components (*Feng et al., 1998*; *Xi et al., 2008*; *Ross et al., 2006*; *Dahle et al., 2010*; *Kato et al., 2002*; *Kim et al., 2011*). Moreover, recent studies in ESCs suggest that NODAL–SMAD2 signaling affects the H3K4Me3 status of target genes through the methyl transferase DPY30 (*Bertero et al., 2015*). However, what dictates the target gene specificity of these collaborating factors has not been determined, nor is it known whether they precede or follow SMAD2 binding to DNA. Furthermore, the sequence of events occurring on chromatin that connects SMAD complex binding to chromatin modifications to target gene regulation is completely unknown.

Here we have taken a genome-wide approach to discover how a dynamic transcriptional program is executed in response to acute and prolonged NODAL/Activin-SMAD2 signaling. We have used the murine P19 embryonic teratoma cell line, which express pluripotency and mesendodermal genes in response to NODAL/Activin signaling. Most significantly, in contrast to ESCs, they do not differentiate when treated with the NODAL/Activin type I receptor inhibitor, SB-431542 (*Vallier et al., 2005*), making them ideal for analyzing transcription from a signal-inhibited baseline. We have performed RNA-sequencing (RNA-seq) and chromatin immunoprecipitation sequencing (ChIP-seq) for SMAD2, two different phosphorylation states of Pol II and various histone modifications in signal-inhibited cells and upon acute and chronic NODAL/Activin signaling. We identify a dynamic transcriptional program downstream of NODAL/Activin signaling which requires a continuous signaling input for its implementation. We go on to define the sequence of events that occur from SMAD2 binding to transcriptional activation, and the mechanisms underlying them. Our work establishes new paradigms for signal-dependent transcriptional regulation.

## Results

### The dynamics of NODAL/Activin signaling in P19 cells

We first assessed pathway activation in P19 cells over time in response to Activin from a signal-inhibited baseline resulting from overnight treatment with SB-431542. We focused on SMAD2 phosphorylation (pSMAD2) as SMAD2 is the predominant receptor-regulated SMAD downstream of NODAL/Activin in these cells (*Figure 1—figure supplement 1A*). Robust induction of pSMAD2 was detected within 30–60 min, which attenuated to lower levels at later time points (*Figure 1A*). Both pSMAD2 induction and target gene expression were partly dependent on Activin dose (*Figure 1—figure supplement 1B and C*). P19s exhibit chronic signaling in the untreated state which results from autocrine ligand production. This is characterized by low levels of pSMAD2 and intermediate levels of target gene expression, and is also evident upon SB-431542 washout (*Figure 1A*, lane 1; *Figure 1—figure supplement 1B and C*, see lanes marked 'media'). The ligands responsible are NODAL and GDF3, which signal via ACVR1B, ACVR2A/B and TDGF1 (*Schier, 2009*) (*Figure 1—figure supplement 2A–C*). Note that as *Tdgf1* expression is controlled by NODAL/Activin signaling and TDGF1 is a very unstable protein, it is rapidly degraded when signaling is terminated (*Figure 1—figure supplement 2D*). As a result P19 cells respond to Activin, but not to NODAL or GDF3, when acutely stimulated from the SB-431542-treated state.

### Activin induces multiple temporal patterns of gene expression

The gene expression profiles (*Figure 1—figure supplement 1C*) demonstrated that different genes are expressed with distinct kinetics, indicating that this cell system has great potential for investigating transcriptional responses to both acute, sustained and chronic NODAL/Activin signaling. To address this on a genome-wide scale we performed RNA-seq in four different conditions: SB-431542-treated cells; 1 hr Activin-treated; 8 hr Activin-treated and untreated (chronic signaling). We identified 747 genes significantly differentially expressed at at least one timepoint relative to the SB-431542-treated state (*Figure 1B*; *Supplementary file 1*). Expression changes for a subset of genes following Activin treatment were validated by qPCR (*Figure 1—figure supplement 3*). Among those

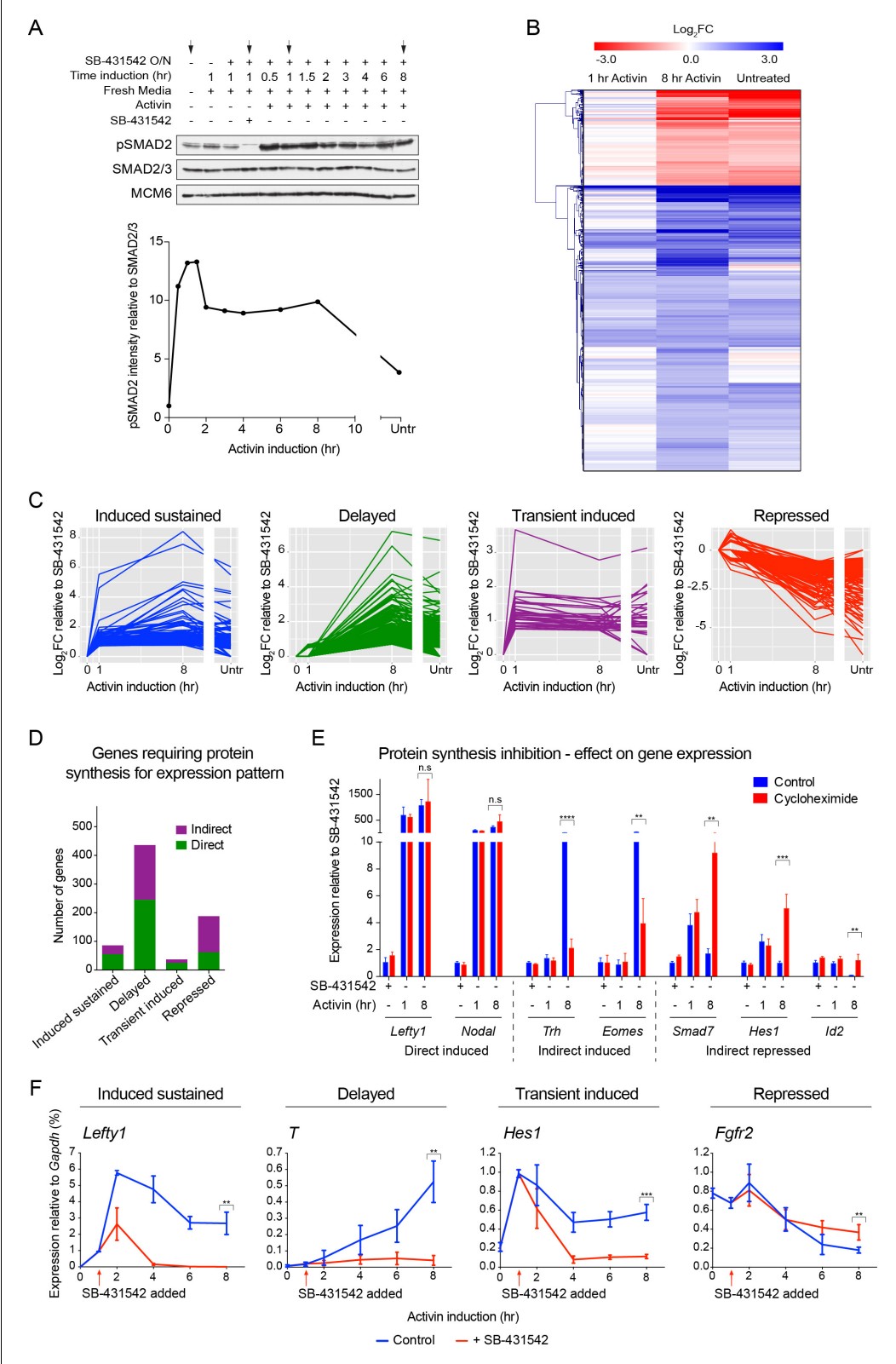

**Figure 1.** The transcriptional response to acute, sustained and chronic NODAL/Activin signaling in P19 cells. (**A**) Western blot showing SMAD2 phosphorylation timecourse upon Activin induction in P19 cells. Cells were treated as indicated and lysates were blotted using the antibodies shown. Arrows indicate the conditions used for the RNA-seq. Below, the phosphorylated SMAD2 (pSMAD2) signal was quantified relative to total SMAD2/3. Untr, untreated (chronic signaling). (**B**) RNA-seq was performed on P19 cells either untreated or incubated overnight with SB-431542, washed out, then

*Figure 1 continued on next page*

*Figure 1 continued*

replaced with full media containing SB-431542 for 1 hr (SB-431542 sample), or with full media containing Activin for 1 hr or 8 hr. Shown is a hierarchically-clustered heatmap of $\log_2$FC values (relative to SB-431542) for each time point for 747 differentially-expressed genes. (C) $\log_2$FC values relative to SB-431542 at each time point plotted for the target genes falling within each of four distinct kinetic clusters. Untr, untreated. (D) Following RNA-seq of control, cycloheximide- or emetine-treated samples in the SB-431542, 1 hr Activin and 8 hr Activin conditions, genes were defined as 'direct' or 'indirect' depending on whether their pattern changed upon protein synthesis inhibition. Displayed are the relative proportions of direct and indirect target genes in each kinetic cluster. (E) Cells were treated with or without cycloheximide as in (D) and processed for qPCR. Transcript levels for a subset of target genes were quantified relative to *Gapdh*. Plotted are the means and SEM of two independent experiments performed in duplicate. n.s., not significant. **** corresponds to a p value of < 0.0001; *** corresponds to a p value of < 0.001 and ** corresponds to a p value of < 0.01. (F) Cells were treated overnight with SB-431542, washed out, then stimulated for the indicated times with Activin (blue line) or SB-431542 was added to the Activin-containing media after 1 hr of Activin treatment (red line). qPCR was performed for the genes shown, which are representatives of each of the four distinct transcriptional profiles. Plotted are the means and SEM of three independent experiments performed in duplicate. *** corresponds to a p value of < 0.001 and ** corresponds to a p value of < 0.01.

The following figure supplements are available for figure 1:

**Figure supplement 1.** Characterization of Activin/NODAL-induced transcription in P19 cells.

**Figure supplement 2.** Characterization of the autocrine signal.

**Figure supplement 3.** Activin/NODAL target genes are induced with different kinetics.

**Figure supplement 4.** Activin/NODAL target gene expression patterns require continuous Activin/NODAL signaling over a sustained time course.

upregulated were genes involved in pluripotency and mesendoderm induction, suggesting an ESC-like signature, and gene ontology analysis identified the enrichment for developmental and TGF-$\beta$ superfamily-regulated processes (*Supplementary file 2*).

To facilitate downstream analysis, we classified the target genes into four categories: 'induced sustained' (acutely upregulated genes whose transcription persists over time); 'delayed' (genes only significantly induced after prolonged signaling); 'transient induced' (acutely upregulated genes whose transcription subsequently declines); 'repressed' (genes that are actively inhibited in response to signaling) (*Figure 1C*). These distinct profiles are not a result of the different half-lives of the mRNAs, as we find no correlation between mRNA half-life and transcription profile (*Figure 1—figure supplement 4A*).

It was evident that, with the exception of the delayed genes, sustained signaling resulted in either repression or dampening of transcription. Even in the 'induced sustained' category where transcript levels were higher after 8 hr of Activin signaling compared with 1 hr, the rate of accumulation of transcripts decreased after 1 hr of Activin stimulation. Consistent with the modulation of gene expression profiles over time, we found that transcription of a substantial proportion of the delayed and repressed genes was perturbed by the presence of protein synthesis inhibitors (*Figure 1D*). A subset of delayed target genes was no longer induced, and many repressed genes, either those inhibited by Activin signaling, or those whose transcription declined following a transient induction, were no longer repressed. These patterns were validated by qPCR (*Figure 1E*).

Extending these results further we investigated whether the distinct profiles of gene expression in response to Activin/NODAL signaling required continuous activity of the pathway. Taking representative examples of target genes in each of the four categories, we found that inhibiting signaling after 1 hr of Activin stimulation with SB-431542 had a dramatic effect on the subsequent transcriptional profile of the target genes. Induced sustained genes require continuous SMAD2 signaling for their long-term transcription, and delayed genes require extended signaling to be induced at all (*Figure 1F*; *Figure 1—figure supplement 4B and C*). For transiently induced genes like *Smad7* and *Hes1* we found that when signaling was terminated after 1 hr their levels fell rapidly to baseline, presumably because of the very short half-life of these mRNAs, whereas in the context of continuous signaling, these genes were repressed in a more gradual fashion (*Figure 1F*; *Figure 1—figure supplement 4*). Transcriptional repression of genes like *Fgfr2* also required on-going signaling

(*Figure 1F*). Thus, modulation of the transcriptional responses over time depends on continuous signaling (see further below).

Taken together these data indicate that Activin stimulation directly induces/represses transcription, and drives a program of secondary events that remodel the transcriptional response over time.

## Integrating RNA-seq with ChIP-seq for SMAD2 and Pol II defines a high confidence Activin-regulated target gene set

To address how Activin induces such a complex program of gene expression we performed ChIP-seq for SMAD2 at the same time points used for the RNA-seq. Activin-induced SMAD2 binding was detected around known direct Activin-responsive genes such as *Lefty1* and *Pmepa1*, thus validating the dataset (*Figure 2A*; *Figure 2—figure supplement 1A,B*). SMAD2 binding for selected target genes was corroborated by ChIP-PCR (*Figure 2—figure supplement 1C*). We next defined peaks of significant SMAD2 enrichment relative to input chromatin (*Zhang et al., 2008*). No peaks were identified in the SB-431542 condition, and 846, 5485 and 2869 SMAD2 peaks were detected at 1 hr Activin, 8 hr Activin and chronic (untreated) signaling states respectively, of which 532 overlap at all three timepoints (*Figure 2B and C*). The majority of binding events occur at regions distal to transcriptional start sites (TSSs), with 70–78% (depending on the condition) identified within 100 kb of an annotated gene (*Figure 2C*).

To link the SMAD2 binding sites (SBSs) to Activin-regulated genes, SMAD2 ChIP-seq data were integrated with the RNA-seq data, along with Pol II ChIP-seq data. For the latter, we performed ChIP-seq for the largest subunit of Pol II specifically phosphorylated on Ser5 or Ser2 of its C-terminal domain (CTD) to differentiate between initiating and elongating forms of Pol II respectively (*Levine, 2011*) (*Figure 2A*; *Figure 2—figure supplement 1D*). By identifying regions of enriched Pol II over gene bodies in stimulated states relative to the SB-431542 state, we defined genes regulated by the pathway. The SMAD2 peaks were associated with the closest regulated gene using a cut-off of 100 kb from an annotated TSS or transcription termination site (TTS). In this way we identified 140 NODAL/Activin transcriptional targets, associated with 478 SMAD2 peaks for further analysis (high confidence dataset; *Supplementary file 1*). A metaprofile of normalized readcounts associated with these peaks is shown in *Figure 2D*. We found that approximately 50% of these SMAD2 peaks were detected within a 10 kb window around the TSS of the regulated gene (*Figure 2E*). The 140 regulated genes were classified into the four different gene expression categories defined above, and we additionally noted that 29 of the induced genes were activated from an undetectable baseline in the SB-431542 state ('baseline off' – see further below) (*Figure 2F*).

To validate the association of SMAD2 peaks with regulated genes we used CRISPR/Cas9 to delete the two major SBSs upstream of *Lefty1* and *Lefty2* (*Figure 2—figure supplement 2*). Deleting the *Lefty1* SBS on both alleles in three individual clones resulted in loss of Activin-induced *Lefty1* expression, without affecting *Lefty2* or other Activin targets (*Figure 2G*; *Figure 2—figure supplement 2*). We obtained the converse when the upstream *Lefty2* SBS was deleted (*Figure 2G*; *Figure 2—figure supplement 2*). Thus we found a one-to-one relationship between SBSs and the genes they regulate, which gave us confidence in our method for associating SMAD2 peaks with target genes.

## Activin/NODAL signaling regulates Pol II via recruitment

Focusing first on the acute transcriptional regulation in response to NODAL/Activin signaling we used the Pol II ChIP-seq analyses to investigate the mechanism by which SMAD2 binding activates transcription. Rapid signal-induced transcription is often characterized by a pause–release mechanism for Pol II activation (*Levine, 2011*). In this scenario, activated Pol II phosphorylated on Ser5 within its CTD is enriched on target genes just downstream of the TSS, associated with pausing factors. Upon active signaling, TF binding induces dissociation of the pausing factors and recruits proteins that positively regulate Pol II elongation, characterized by Ser2 phosphorylation (*Adelman and Lis, 2012*).

Metaprofiles were used to visualize enrichment of initiating Pol II (Ser5P) and elongating Pol II (Ser2P) on all genes and they showed the characteristic patterns, peaking at the TSS and TTS respectively (*Descostes et al., 2014*) (*Figure 3A*). We noted that the relative changes in Pol II Ser5P and Pol II Ser2P for the differentially-regulated target genes followed similar patterns (*Figure 3B*), and

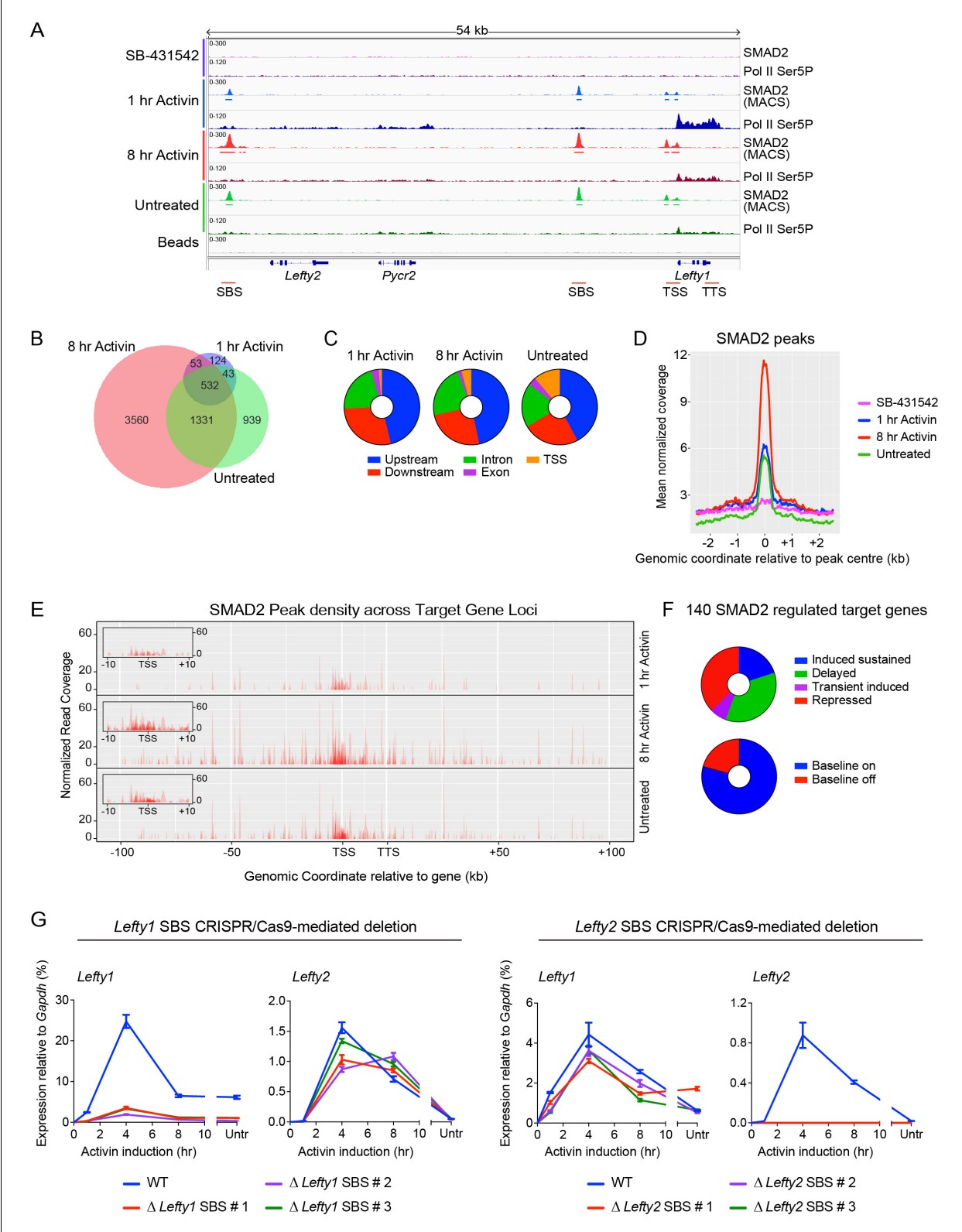

**Figure 2.** Combining RNA-seq with ChIP-seq for SMAD2 and Pol II leads to the definition of a high confidence dataset. (**A**) IGV browser display of the *Lefty1/Lefty2* genomic locus, showing tracks and MACS-called peaks for SMAD2 and tracks for Pol II Ser5P. Red lines below are regions for which ChIP-qPCR primers were designed. (**B**) SMAD2 ChIP-sequencing was performed on P19 cells in the same conditions as the RNA-seq (*Figure 1*). The numbers of peaks in each condition are shown in the Venn diagram. (**C**) Distribution of SMAD2 peaks based on their genomic annotation to upstream or

*Figure 2 continued*

downstream intergenic regions, TSS (considered TSS ± 500 bp), exonic or intronic location. (D) Metaprofiles for the SMAD2 peaks in the high confidence dataset. Normalized read counts centred on the peak ± 2 kb are shown for the four different experimental conditions. (E) Plots displaying the density of SMAD2 peaks in the high confidence dataset and their distance from the annotated TSS of the nearest regulated target gene within a ± 100 kb window for each condition. All genes are modeled as 20 kb from TSS to TTS. Insets show density of SMAD2 peaks ±10 kb centred around the TSS of the modeled target genes. (F) Distribution of the high confidence dataset of 140 SMAD2-regulated genes within each of the four categories, or segregated according to 'baseline on' or 'baseline off' in the SB-431542 state. (G) Left panel: Deletion of the upstream *Lefty1* SBS in three independent clones inhibits Activin-induced *Lefty1* transcription, but not *Lefty2* transcription. Right panel. Deletion of the upstream *Lefty2* SBS in three independent clones inhibits Activin-induced *Lefty2* transcription, but not *Lefty1* transcription. Transcript levels were quantified relative to *Gapdh*. A representative experiment (means ± SD) is shown. Untr, untreated.

The following figure supplements are available for figure 2:

**Figure supplement 1.** Characterization of SMAD2 and Pol II binding in response to Activin signaling.

**Figure supplement 2.** CRISPR/Cas9-induced deletion of the *Lefty1* and *Lefty2* upstream SBSs.

the Ser2P and Ser5P values correlated well (*Figure 3C*). These observations suggested that Pol II recruitment is the dominant mechanism for transcriptional activation in response to Activin, because for a pause–release mechanism we would expect Pol II Ser5P and Pol II Ser2P to be differentially enriched (*Sawicka et al., 2014*). We addressed this directly using metaprofiles to examine the relative enrichment of Pol II Ser5P and Pol II Ser2P around the TSSs of target genes subdivided with respect to their kinetic expression pattern. Most importantly there was no paused Pol II Ser5P at the TSS in the SB-431542-treated state for the genes with a silent baseline ('baseline off') nor any enrichment of Pol II Ser2P in these conditions (*Figure 3D*). For the 'baseline on' genes, where a peak of Pol II Ser5P *was* seen at the TSS in the SB-431542-treated state, this was accompanied by enrichment of Pol II Ser2P throughout the body of the gene and a peak at the TTS, indicating on-going transcription of these genes as expected (*Figure 3D*). Moreover, the shape of the profiles did not change with ligand induction; the amplitude simply increased, which reflects changes in Pol II recruitment. For the other categories, the enrichment of Pol II Ser5P generally followed the gene expression dynamics (*Figure 3D*).

Taking all these data together we conclude that Activin/NODAL signaling regulates transcription of target genes in response to signaling via de novo recruitment of Pol II, rather than via pause–release.

## Localized SMAD2 chromatin binding does not directly correlate with transcription over prolonged time courses

We next investigated how SMAD2 regulates the long-term transcriptional responses to Activin signaling, in particular determining whether transcriptional repression correlated with loss of SMAD2 binding. This is a crucial question in the light of the results above showing that modulation of the transcriptional profiles over time requires on-going Activin signaling. To quantitate SMAD2 binding for each gene we defined a SMAD2 footprint that incorporates both peak intensity and peak number associated with the regulated gene. We compared how the SMAD2 footprint changed over time relative to transcription as measured by RNA-seq and Pol II occupancy (*Figure 4A and B*). Although the majority of directly induced genes (induced sustained or transient induced) showed highest Pol II occupancy after 1 hr Activin signaling, mirroring pSMAD2 activation, SMAD2 binding did not correlate with the transcriptional kinetics (*Figure 4*; *Figure 4—figure supplement 1*). In many cases SMAD2 chromatin occupancy increased around genes after prolonged signal exposure when Pol II occupancy was downregulated (*Figure 4A* – see the transient induced and repressed gene categories). Two very clear examples of this phenomenon are *Smad7* and *Tbx3* (*Figure 4B*). This strongly suggests that secondary repression of transcription in response to Activin is not a result of loss of SMAD2 binding, but rather subsequent recruitment of repressors by chromatin-bound SMAD2.

We therefore conclude that the requirement for continuous Activin signaling for modulation of long term responses is reflected in the continued presence of SMAD2 at regulatory regions of target genes. As a result, SMAD2 binding does not correlate with transcriptional kinetics.

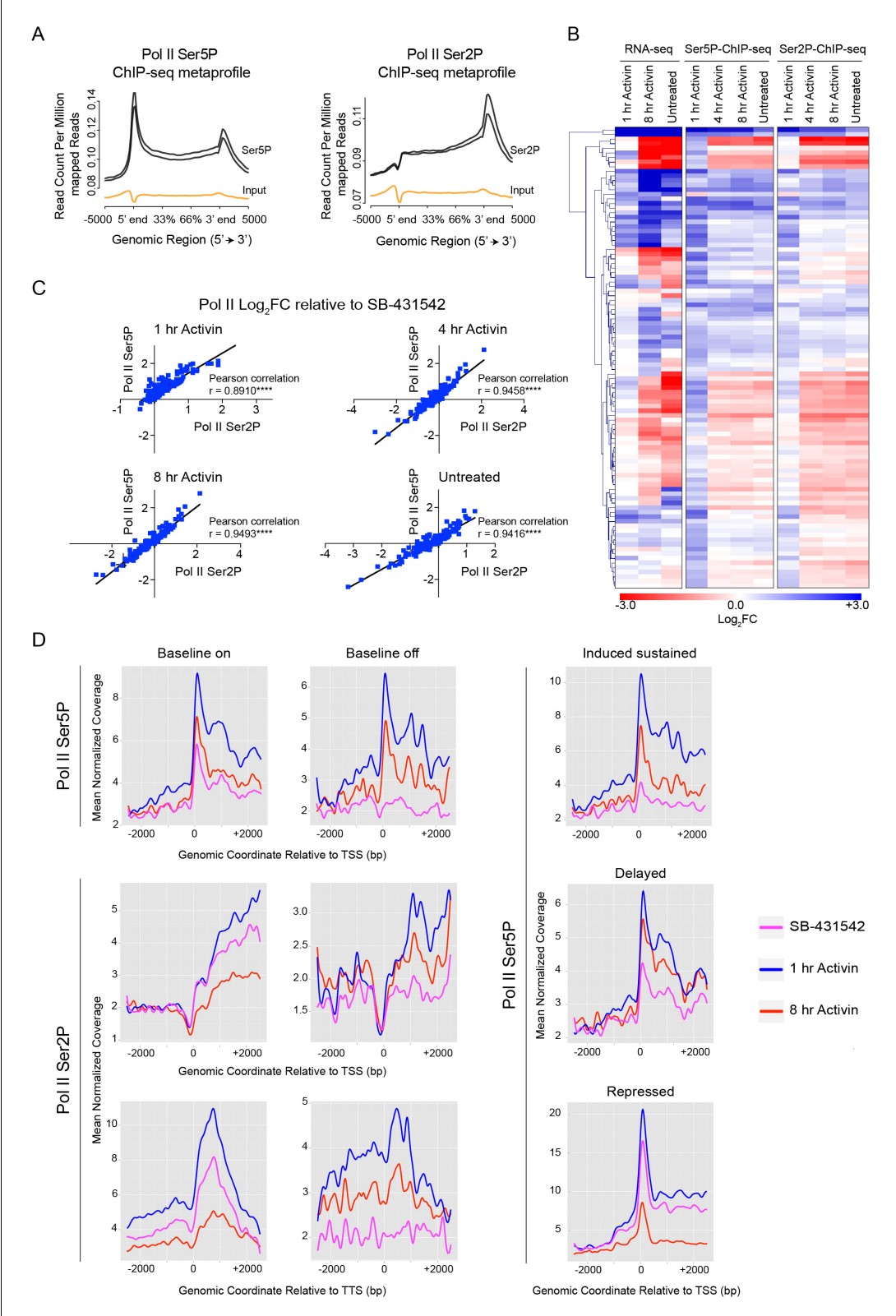

**Figure 3.** Activin-SMAD2 signaling regulates Pol II via de novo recruitment. (**A**) Average metaprofiles for each of two replicates from the 1 hr Activin sample of the ChIP-seq for Ser5P or Ser2P isoforms of Pol II. Normalized read count across all genes is shown. The orange line denotes the input. (**B**) Activin-SMAD2 target genes from the high confidence dataset showing differential Pol II binding were hierarchically-clustered and the log₂FC relative to SB-431542 for mean normalized read depth for Pol II Ser 5P and Pol II Ser2P is displayed in a side-by-side comparison with the transcript levels

*Figure 3 continued on next page*

*Figure 3 continued*

(determined by RNA-seq). (**C**) Correlations of the log₂FC values of affected target genes for Pol II Ser2P and Ser5P for each individual time point. (**D**) Metaprofiles generated for Pol II Ser5P or Pol II Ser2P ChIP-seq data across a 5 kb window surrounding the TSS or the TTS for indicated subsets of target genes. Each colored line traces the normalized read depth of Pol II Ser5P or Pol II Ser2P in each condition.

## SMAD2 induces changes in the chromatin landscape to regulate transcription

SMADs are known to activate transcription through chromatin remodeling, rather than by recruiting the basal transcription machinery (*Ross et al., 2006*). However, it is not known to what extent the SMADs directly induce changes in the chromatin landscape at target binding sites, and if they do, with what dynamics these occur. To answer this question we performed ChIP-seq for histone modifications in the same conditions used for the Pol II and SMAD2 ChIP-seq analyses. We analysed opposing activating and repressive histone modifications, namely H3K9Ac and H3K9Me3, as well as H3K27Ac and H3K27Me3 (*Calo and Wysocka, 2013*; *Zentner and Scacheri, 2012*). ChIP-seq for total H3 was also carried out to identify regions of nucleosome depletion.

No significant enrichment was found for the repressive marks H3K27Me3 or H3K9Me3 at SBSs, or over the Activin-regulated target genes, although it was enriched at other loci (data not shown).

In contrast, histone H3 acetylation was observed around the majority of SMAD2 peaks, and in many cases the signal increased in response to Activin signaling (*Supplementary file 1*; *Figure 5A*; *Figure 5—figure supplement 1*; *Figure 5—figure supplement 2*). As expected, H3 acetylation was also enriched over the gene bodies of the Activin-induced target genes (*Figure 5A*; *Figure 5—figure supplement 2*; *Figure 5—figure supplement 3*). Acetylation over the TSS generally followed the kinetics of gene expression (*Figure 5—figure supplement 3*).

To assess to what extent NODAL/Activin signaling induced changes in histone modification and/ or occupancy around SBSs, we generated metaprofiles for a 5 kb window centered around all loci in the high confidence dataset that have a SMAD2 peak in at least one signaling condition. When observing all SBSs together we found that they were generally nucleosome-depleted and ligand-induced histone acetylation was restricted to around 1.5 kb either side of the SBS (*Figure 5B*, left-hand panels). The local acetylation is also evident in the IGV browser displays (*Figure 5A*; *Figure 5—figure supplement 2*). The two H3Ac modifications were similarly induced upon acute stimulation, and ligand-induced enrichment around target sites correlated well (*Figure 5C*). Lysines 9 and 27 of H3 are not the only residues acetylated in response to signaling, as we also detected significant enrichment of H3K18 and H3K23 acetylation in response to Activin on the nucleosomes surrounding the *Lefty1* SBS, which was used as a representative example (*Figure 5D*). A good candidate for inducing H3 acetylation is EP300 (*Ross et al., 2006*), as it is enriched over SBSs in response to acute and sustained signaling with similar kinetics to SMAD2 (*Figure 5E*).

We next classified the SMAD2 peaks according to the kinetic profiles of the target genes, and a much more dramatic effect of signaling on histone modification/occupancy was revealed. For 'baseline off' genes, a robust ligand-induced decrease in H3 over SBSs was observed after 1 hr Activin stimulation, and a concomitant increase was seen in H3K9Ac and H3K27Ac at either side of the SMAD2 peak (*Figure 5B*). These effects were even more pronounced if we focused only on those sites which showed the highest induction of histone acetylation (see *Figure 6D* for definition of these peaks). This revealed a greater than two fold decrease in H3 occupancy at SBSs with ligand stimulation and an induction of histone acetylation from a flat baseline (*Figure 5B*, right-hand panels). Thus, SMAD2 binding in response to acute Activin signaling to transcriptionally inactive chromatin results in nucleosome displacement and induction of histone acetylation at adjacent nucleosomes.

The Activin-induced decrease in histone H3 over SBSs for 'baseline off' genes, and the increase in H3 acetylation, were verified for a subset of genes by ChIP-PCR and FAIRE-PCR. The chromatin of the *Lefty1* and *Pmepa1* SBSs was in a closed conformation in the SB-431542 condition with H3 present and became depleted in H3 and more accessible after 1 hr Activin treatment (*Figure 5A,F*; *Figure 5—figure supplement 1*; *Figure 5—figure supplement 2*). In contrast, SBSs of genes transcribed in the absence of signaling (*Pou5f1*, *Trh*) were in an open conformation that did not change following 1 hr Activin treatment (*Figure 5F*; *Figure 5—figure supplement 1*).

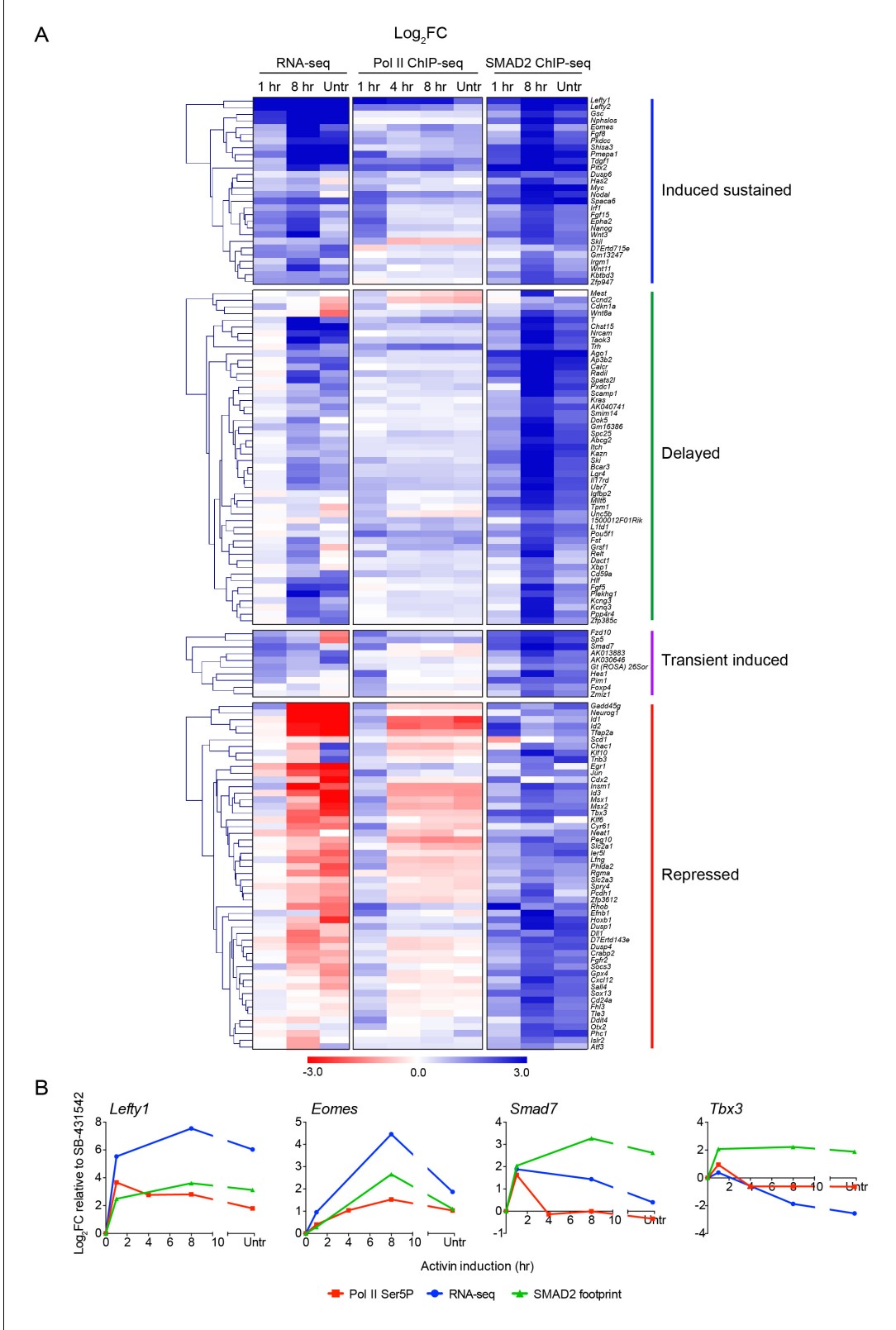

**Figure 4.** Localized SMAD2 chromatin binding does not directly correlate with transcription. (**A**) Hierarchically-clustered heatmaps for each of the four different kinetic groups of target genes showing log₂FC values relative to SB-431542 for gene expression as determined by RNA-seq (left), mean normalized read depth for Pol II Ser5P (middle) and SMAD2 footprint (right). Untr, untreated. (**B**) Transcriptional dynamics of different target genes (*Lefty1*, *Eomes*, *Smad7*, *Tbx3*). (For IGV browser displays, see *Figure 2A* and *Figure 4—figure supplement 1*). Plotted are quantifications of transcript

*Figure 4 continued*

levels obtained by RNA-seq, Pol II Ser5P occupancy expressed as mean normalized read depth, and SMAD2 footprint, all depicted as log$_2$FC relative to SB-431542. Untr, untreated.

The following figure supplement is available for figure 4:

**Figure supplement 1.** IGV browser displays of genes and associated peaks showing differential transcription and SMAD2 binding dynamics.

Thus, activated SMAD2 has two distinct modes of binding. It targets sites in closed chromatin devoid of histone acetylation, whereupon SMAD2 binding induces nucleosome release and acetylation of adjacent nucleosomes. At other loci SMAD2 recognizes already nucleosome-depleted sites, flanked by acetylated histones, and induces further acetylation.

## SMAD2 binding correlates with histone acetylation

To understand how SMAD2 finds its binding sites in chromatin we investigated whether there was any correlation between SMAD2 binding and histone acetylation. To this end we quantified H3K27Ac and H3K9Ac read depth in 5 kb windows surrounding all loci in the high confidence dataset with a SMAD2 peak in at least one signaling condition. Histone acetylation is highly variable in the SB-431542 state and also changes over time at many SBSs (*Figure 6A*). We found that overall SMAD2 binding is stronger around those loci defined by high H3K9 and H3K27 acetylation (*Figure 6B*). Moreover, the acetylation status in the SB-431542-treated state predicted SMAD2 binding for the 1 hr peaks, but not for the 8 hr peaks (*Figure 6C*). Because a subset of SMAD2 binding events occur at non-acetylated chromatin (see above), we also investigated whether higher induction of acetylation at these sites following signal activation was associated with higher enrichment of SMAD2. Indeed, SMAD2 normalized read depth was significantly higher at regions where histone acetylation is induced (*Figure 6D*).

Taken together with the previous section, these results clearly indicated that SMAD2 binds unacetylated chromatin. This then raised the question of whether there is an additional 'landmark' that specifies SMAD2 target sites. We hypothesized that H3K4 monomethylation would be a good candidate, as it is known to define poised or primed enhancer sites (*Calo and Wysocka, 2013*). Using the major upstream SMAD2 peak at the *Lefty1* locus as an example of an SBS that is neither acetylated in the absence of signaling nor nucleosome-depleted (*Figure 5*), we observed that H3K4Me1 is enriched prior to signal activation, and persisted with sustained NODAL/Activin signaling (*Figure 6E*).

Thus, SMAD2 binding strength correlates with histone acetylation status, either in a constitutive or acute signal-induced manner and H3K4Me1 likely precedes SMAD2 binding and H3 acetylation.

## Sequence-specific TFs are required for induction of SMAD2 target genes

As SMAD2 cannot bind DNA itself, it requires additional TFs for recruitment to chromatin (*Gaarenstroom and Hill, 2014*). We therefore examined the sequences under the summits of the SMAD2 peaks in the high confidence dataset to identify binding sites for recruiting TFs. As expected we found sites for known SMAD2 co-factors, including FOXH1, POU5F1, and also the SMAD3/4 binding element, which reflects DNA binding of SMAD4 in the activated SMAD complex (*Inman and Hill, 2002*) (*Figure 7—figure supplement 1A*). Novel motifs were also found, as well as a weaker enrichment for the NANOG, EOMES and TEAD sites (*Figure 7—figure supplement 1A and B*).

We confirmed the requirement of FOXH1 and POU5F1 for Activin-induced transcription for some representative target genes, which have at least one SMAD2 peak containing binding sites for these TFs (*Figure 7A*; *Figure 7—figure supplement 2A*). The effect of FOXH1/POU5F1 knockdown was gene-dependent. *Lefty1* and *Eomes* required both FOXH1 and POU5F1, whilst *Tdgf1* was dependent only on POU5F1, and *Pmepa1* did not require either, consistent with the fact that it is induced by TGF-$\beta$ in many different cell types (*Levy and Hill, 2005*; *Watanabe et al., 2010*). Similarly, we tested the requirement for NANOG, EOMES or TEAD on the transcription of target genes containing the relevant motif under their SMAD2 peaks. However, knockdown of these factors had no

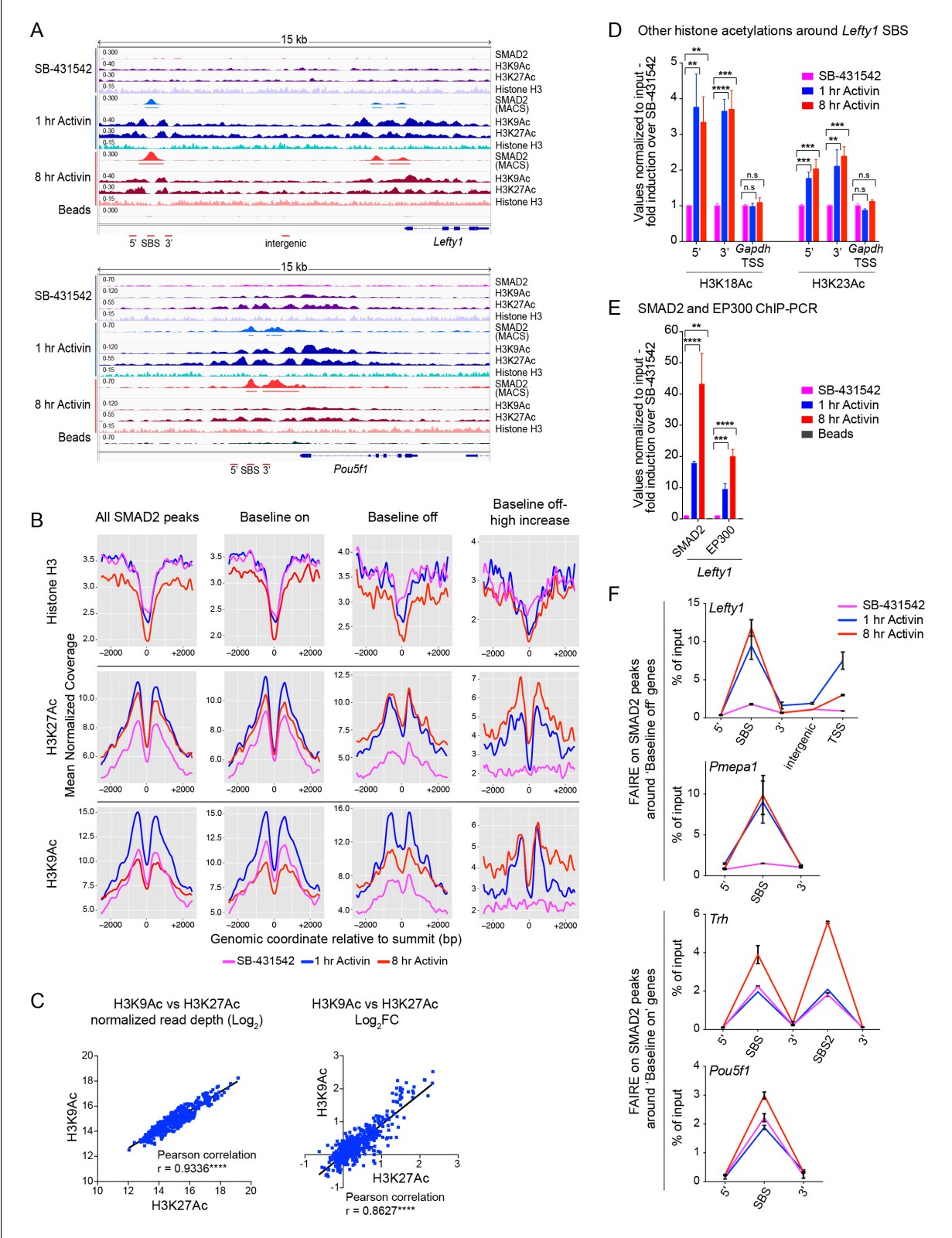

**Figure 5.** Changes in chromatin landscape around SBSs in response to signaling. (**A**) IGV browser displays for the *Lefty1* and *Pou5f1* loci displaying ChIP-seq tracks for SMAD2, H3K9Ac, H3K27Ac and total histone H3 for the indicated treatments. For the SMAD2 ChIP-seq the MACS-called peaks are also shown. (**B**) Metaprofiles denoting coverage of total histone H3, H3K27Ac and H3K9Ac in a 5 kb window surrounding the average summit across SMAD2 consensus peaks. Each line within a plot traces the normalized read depth of H3 or acetylated H3 for each treatment. The left panels show

*Figure 5 continued on next page*

*Figure 5 continued*

metaprofiles for all 478 SMAD2 peaks, while the other panels only show those regions found near genes which are induced from an 'on' baseline or from an 'off' baseline. The right panels show metaprofiles for the subset of SMAD2 peaks which is associated with genes induced from an 'off' baseline and show changes in H3Ac enrichment from a low baseline as defined in *Figure 6D*. (C) Correlation plots showing normalized read depth (Log$_2$) (left graph) or log$_2$FC relative to SB-431542 (right graph) for H3K27Ac and H3K9Ac values, detected in the 1 hr Activin sample. (D) ChIP-PCR on the nucleosomes flanking the *Lefty1* SBS (A) for H3K18Ac and H3K23Ac in P19 cells treated as indicated. Plotted are the means and SEM of three independent experiments. (E) ChIP-PCR showing enrichment of both SMAD2 and EP300 at the *Lefty1* SBS in P19s treated as indicated. Plotted are the means and SEM of four independent experiments. (F) FAIRE-ChIP was performed on SB-431542, 1 hr Activin or 8 hr Activin-treated P19 cells. Soluble chromatin fractions were assayed for enrichment of the specific regions indicated which are highlighted in the IGV browser displays in panel (A) and *Figure 2—figure supplement 1* and *Figure 5—figure supplement 1*. A representative experiment in shown (means ± SD). See *Figure 5—figure supplement 1B* for the averages of the three experiments and the statistical analyses. In D and E, n.s., not significant; **** corresponds to a p value of < 0.0001; *** corresponds to a p value of < 0.001 and ** corresponds to a p value of < 0.01.

The following source data and figure supplements are available for figure 5:

**Source data 1.** H3K27Ac and H3K9Ac values detected in the 1 hr Activin sample.

**Figure supplement 1.** Validation of induced acetylation and FAIRE at and around SBSs.

**Figure supplement 2.** IGV browser displays of genes and associated peaks showing changes in chromatin landscape.

**Figure supplement 3.** Correlation between histone H3 acetylation changes and gene expression.

effect, suggesting that they may not be essential for Activin-induced transcription in these cells (data not shown).

## Binding of FOXH1-SMAD2 complexes is required for chromatin remodeling at a subset of target genes

We noted that the FOXH1 motif is centrally enriched relative to the SMAD2 peak summit, compared with the POU5F1 motif, which is more evenly distributed (*Figure 7B*). This suggested that these two TFs may play fundamentally different roles in SMAD2 binding, with FOXH1 directly recruiting SMAD2, whilst POU5F1 having a more auxillary role.

Strikingly, SMAD2 binding strength correlated with the presence of a FOXH1 motif and the FOXH1 binding site was significantly enriched in SBSs bound by SMAD2 after 1 hr Activin stimulation, compared with 8 hr (*Figure 7—figure supplement 1C,D*). Since FOXH1 is a member of the forkhead transcription factor family which frequently act as pioneer factors (*Zaret and Carroll, 2011*), we hypothesized that FOXH1 may act as such to recruit SMAD2 to sites of closed unacetylated chromatin. We raised an antibody against mouse FOXH1 (*Figure 7—figure supplement 3*) and performed ChIP-PCR on a subset of SBSs. We found no binding of FOXH1 at all at any SBS in the SB-431542-treated cells, but binding was rapidly and strongly inducible upon Activin stimulation at the SBSs for *Lefty1*, *Lefty2*, and *Pitx2*, which are genes whose transcription is FOXH1-dependent (*Figure 7C*). There was little or no inducible binding at the SBSs flanking *Pmepa1* and *Pou5f1*, as expected, since expression of these genes does not require FOXH1 (*Figure 7C*). The same results were obtained using a P19 cell line that stably expresses near-endogenous levels of MYC-tagged FOXH1 and performing ChIP-PCR with an anti-MYC antibody (*Figure 7—figure supplement 2B–D*). Thus for genes that are regulated by FOXH1, it is recruited to DNA together with SMAD2 upon Activin stimulation, and does not act as a classic pioneer factor.

We next tested whether FOXH1 is directly required for SMAD2 binding to target loci and for the subsequent changes in chromatin landscape in response to signaling. Following knockdown of FOXH1, Activin-induced recruitment of SMAD2 was inhibited at SBSs around *Lefty1*, *Lefty2* and *Pitx2* (*Figure 7D*; *Figure 7—figure supplement 2E*), and Activin-induced H3K27 acetylation at the *Lefty1* and *Lefty2* sites was also perturbed (*Figure 7E*; *Figure 7—figure supplement 2F*; *Figure 7—figure supplement 4*). As expected, recruitment of SMAD2 and subsequent H3 acetylation at sites around genes shown not to be dependent on FOXH1 such as *Tdgf1*, *Pmepa1*, *Pou5f1* and *Smad7* was unaffected (*Figure 7D,E*; *Figure 7—figure supplement 4*). We also investigated whether

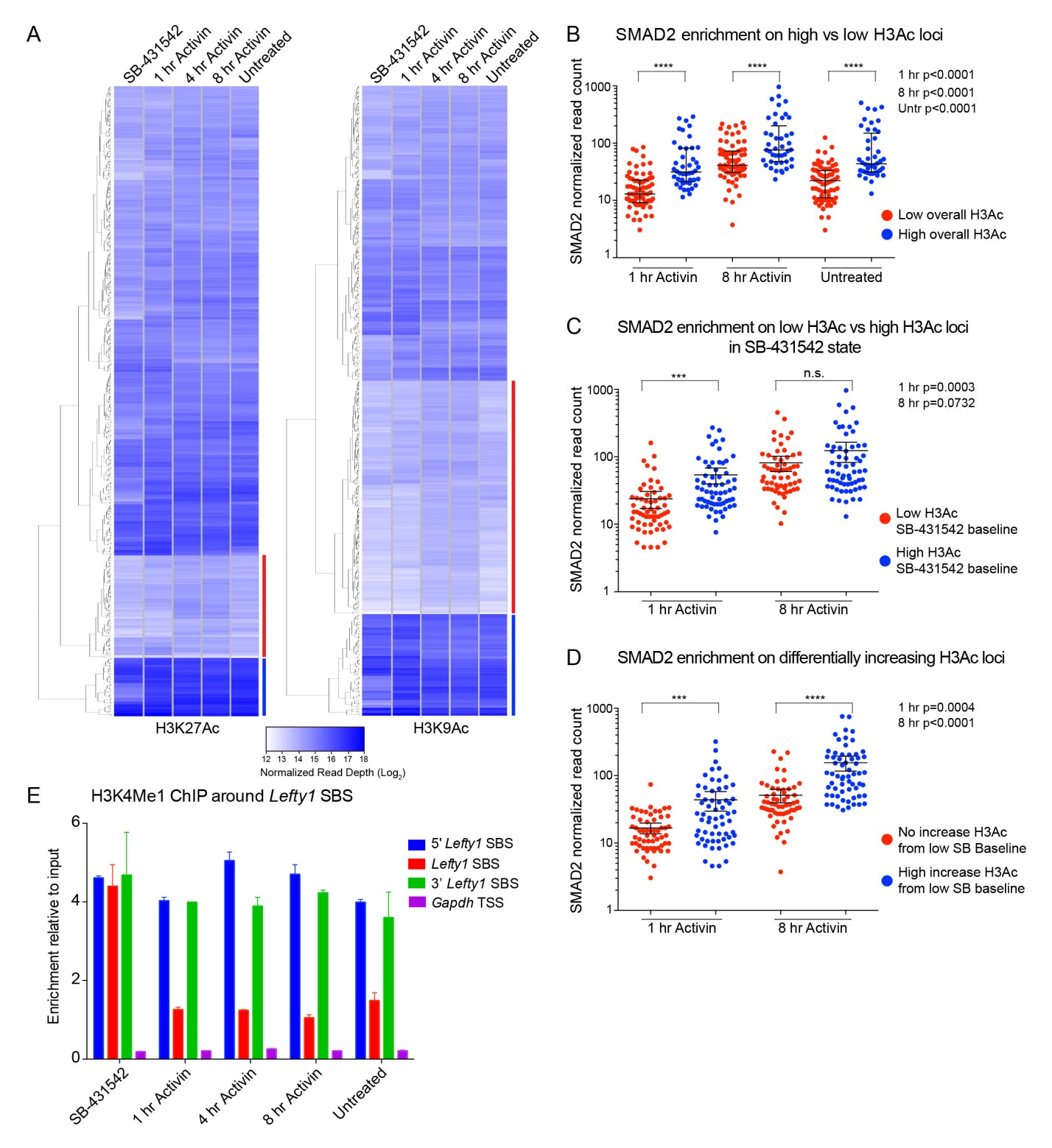

**Figure 6.** Relationship between chromatin state and SMAD2 binding strength. (**A**) Hierarchical clustering of average read intensity (Log₂) over a 5 kb window surrounding all SMAD2 consensus peak summits for each indicated treatment is shown for H3K27Ac (left panel) and H3K9Ac (right panel). Red line, peaks associated with low overall acetylation; blue line, peaks found in highly acetylated chromatin. (**B–D**) Correlation between normalized SMAD2 read counts and H3 acetylation state. The SMAD2 read counts over consensus peaks in either the 'low overall acetylation' (red) or 'high overall acetylation' (blue) category were plotted for the three conditions (**B**). The SMAD2 read counts over consensus peaks that were defined as either low

*Figure 6 continued on next page*

*Figure 6 continued*

H3Ac or high H3Ac in the SB-431542 state were plotted for the 1 hr and 8 hr Activin conditions (**C**). The SMAD2 read counts over subsets of loci selected based on changes in H3Ac enrichment from a low baseline (**D**). In all cases the black bars indicate the mean and 95% confidence interval. n.s., not significant. The p-values are given in the plots. (**E**) ChIP-PCR for H3K4Me1 for the nucleosomes at and flanking the *Lefty1* SBS or a control region within the *Gapdh* coding locus. A representative experiment is shown (means ± SD).

The following source data is available for figure 6:

**Source data 1.** Average read intensity over a 5 kb window surrounding all SMAD2 consensus peak summits for H3K27Ac and H3K9Ac.

FOXH1 is required for signal-dependent nucleosome displacement using FAIRE-PCR. Activin-induced chromatin accessibility of the *Lefty1* SBS was substantially inhibited when FOXH1 was depleted (*Figure 7F*; *Figure 7—figure supplement 4*), indicating that the FOXH1–SMAD2 complex is required there for nucleosome eviction. FOXH1 was not required for ligand-induced accessibility of the *Pmepa1* SBS, consistent with the lack of FOXH1 dependence for SMAD2 binding at this site or for *Pmepa1* transcription (*Figure 7F*; *Figure 7—figure supplement 4*). The constitutively open *Pou5f1* SBS was also unaffected by depletion of FOXH1, as expected (*Figure 7F*; *Figure 7—figure supplement 4*).

FOXH1 is therefore responsible for directing SMAD2 to a subset of target sites in response to acute signaling. This in turn leads to nucleosome depletion and histone acetylation. FOXH1 does not act as a pioneer factor, but binds together with activated SMAD2.

## SMARCA4 is required for nucleosome eviction at a subset of SMAD2 binding sites

Having demonstrated that activated SMAD2 binds to closed chromatin and that it induces nucleosome eviction, we dissected the underlying mechanism. Previous work has shown that SMARCA4, the ATP-dependent helicase component of the SWI-SNF nucleosome remodeling complex (*Lange et al., 2011*), is required for TGF-β-induced transcription of many, though not all, target genes (*Ross et al., 2006*; *Xi et al., 2008*). At what level it functions has not been clear. As SMARCA4 is known to interact with activated SMAD2 (*Ross et al., 2006*; *Xi et al., 2008*), we hypothesized that it was a good candidate for promoting chromatin remodeling and/or nucleosome eviction at SMAD2 binding sites. To address this possibility we first tested the effect of SMARCA4 knockdown on transcription of a selection of Activin target genes. Activin-induced transcription of three 'baseline off' genes, *Lefty1*, *Lefty2* and *Pmepa1,* was inhibited by depletion of SMARCA4, whilst the Activin-induced transcription of two 'baseline on' genes, *Smad7* and *Pitx2* was unaffected (*Figure 8A*). The effect of SMARCA4 depletion on SMAD2 chromatin binding correlated well with the transcription assays. Recruitment of SMAD2 to the SBSs flanking *Lefty1*, *Lefty2* and *Pmepa1* required SMARCA4, but SMAD2 recruitment to those regulating *Pitx2* and *Smad7* was not substantially affected (*Figure 8B*). Consistent with the idea that SMAD2 binding leads to flanking histone acetylation and nucleosome eviction, we found that Activin-induced H3K27 acetylation and increased chromatin accessibility at the SBSs flanking *Lefty1*, *Lefty2* and *Pmepa1* were also dependent on SMARCA4 (*Figure 8C,D*; *Figure 8—figure supplement 1*). In contrast, Activin-induced acetylation and accessibility of the *Smad7* and *Pitx2* SBSs was markedly less dependent on SMARCA4 (*Figure 8C,D*; *Figure 8—figure supplement 1*).

We conclude therefore that SMARCA4 is required for SMAD2 binding to closed chromatin at 'baseline off' genes, where it mediates nucleosome eviction and flanking acetylation.

## Discussion

### Delineating the sequence of events from SMAD2 binding to transcriptional regulation

By combining RNA-seq with ChIP-seq for SMAD2, two distinct forms of Pol II, histone H3 acetylations and total H3 in P19 embryonic carcinoma cells, we have addressed how a single pathway induces a program of gene expression that is constantly remodeled over time. Our work has determined

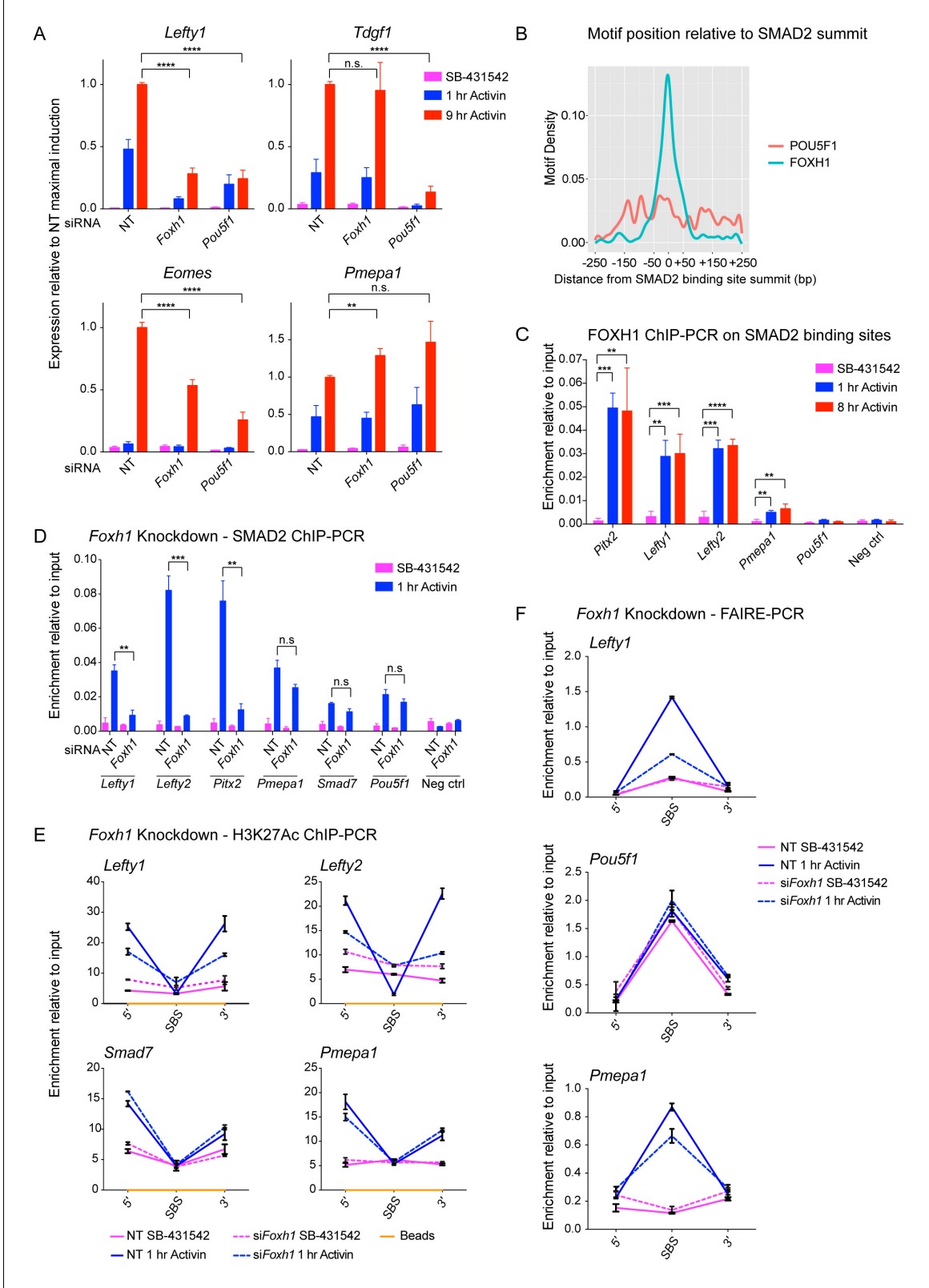

**Figure 7.** FOXH1 is required for SMAD2 recruitment and nucleosome remodeling at a subset of Activin target genes. (**A**) P19 cells were transfected with siRNAs directed against either *Pou5f1* or *Foxh1*, along with a non-targeting control (NT). Following signal inhibition or Activin induction, qPCR was performed for the genes shown. Plotted are the means and SEM of three independent experiments performed in duplicate of gene expression values normalized to endogenous *Gapdh* values. (**B**) The position of FOXH1 or POU5F1 motifs (regardless of orientation or strand) was plotted relative to the

*Figure 7 continued on next page*

*Figure 7 continued*

summit of all the peaks used in *Figure 7—figure supplement 1A*. (C) ChIP-PCR for FOXH1 at the indicated SBSs and negative control (Neg ctrl) region using P19 cells in the conditions shown. Plotted are the means and SEM of two independent experiments performed in duplicate. (D–F) P19 cells were transfected with either non-targeting (NT) or *Foxh1* siRNAs, which were then signal inhibited (SB-431542) or stimulated with Activin for 1 hr after SB-431542 washout. They were assayed for SMAD2 ChIP-PCR (D), H3K27Ac ChIP-PCR (E) or FAIRE-PCR (F) on the selected SBSs indicated. Plotted in D are the means and SEM of three independent experiments. Neg ctrl, negative control. The data in E and F are from a representative experiment of three (means ± SD). See *Figure 7—figure supplement 4* for the averages of the three experiments and the statistical analyses. In A, C and D, n.s., not significant; **** corresponds to a p value of < 0.0001; *** corresponds to p value of < 0.001; ** corresponds to a p value of < 0.01.

The following source data and figure supplements are available for figure 7:

**Source data 1.** The position of FOXH1 or POU5F1 motifs relative to the summit of all the consensus peaks.

**Figure supplement 1.** TF binding sites under SMAD2 peaks.

**Figure supplement 2.** The role of FOXH1 in SMAD2-mediated transcription.

**Figure supplement 3.** Characterization of the in-house anti FOXH1 antibody.

**Figure supplement 4.** The role of FOXH1 in SMAD2-mediated chromatin remodeling.

the sequence of events that occur from SMAD2 recruitment to chromatin to transcriptional activation and establishes new paradigms for ligand-induced SMAD2-dependent transcription (*Figure 9*).

First, we have shown that activated SMAD2 exhibits two distinct modes of binding. For a subset of genes SMAD2 binds in response to acute Activin signaling to transcriptionally silent chromatin. These sites appear to be marked by H3K4Me1 and SMAD2 binds them without the requirement for a pioneer factor. In fact we show that SMAD2 and the co-activator FOXH1 both bind inducibly upon Activin signaling. SMAD2 binding induces nucleosome displacement and histone H3 acetylation at multiple sites at adjacent nucleosomes. The acetylation is likely induced by EP300 as we demonstrate that this HAT is recruited to chromatin with the same kinetics as SMAD2. Moreover, SMAD2 recruitment, histone acetylation and chromatin accessibility requires SMARCA4. At other targets, which have a detectable level of basal transcription, SMAD2 binds to pre-existing nucleosome-depleted sites that are already flanked by acetylated H3-containing nucleosomes. H3 acetylation is further increased upon SMAD2 binding at these sites. Following SMAD2 binding and induction of local histone acetylation, transcription is initiated by inducing Pol II recruitment, rather than via a pause–release mechanism commonly associated with signal-induced transcription. Pol II recruitment correlates with further histone acetylation both at the TSS and in the body of the regulated gene. Finally, we have established that the long-term dynamic transcriptional profiles downstream of NODAL/Activin require a continuous signaling input. This is evident at the level of chromatin since we find that SMAD2 remains bound at enhancers at prolonged times after Activin induction even though target gene transcription is frequently attenuated or completely terminated. Thus SMAD2 binding does not correlate directly with transcription of target genes, and our results strongly suggest that chromatin-bound SMAD2 sequentially recruits multiple co-factors that modulate the transcriptional output during prolonged signaling.

## Dynamics of SMAD2-mediated transcription in response to acute and prolonged signaling

We identified four distinct kinetic patterns of gene expression induced by Activin. Approximately a third of the target genes require protein synthesis for their correct induction/repression in response to signaling, demonstrating that the initial response is remodeled at later times by factors that are themselves encoded by Activin target genes. Furthermore, as the kinetic patterns of gene transcription require continuous Activin signaling, many of these factors likely collaborate with pSMAD2.

The increased numbers and/or intensity of SMAD2 peaks observed during prolonged signaling suggest that more SMAD2-containing complexes become stably associated with chromatin as a result of SMAD2-mediated transcription. In some cases the additional peaks occur at target genes

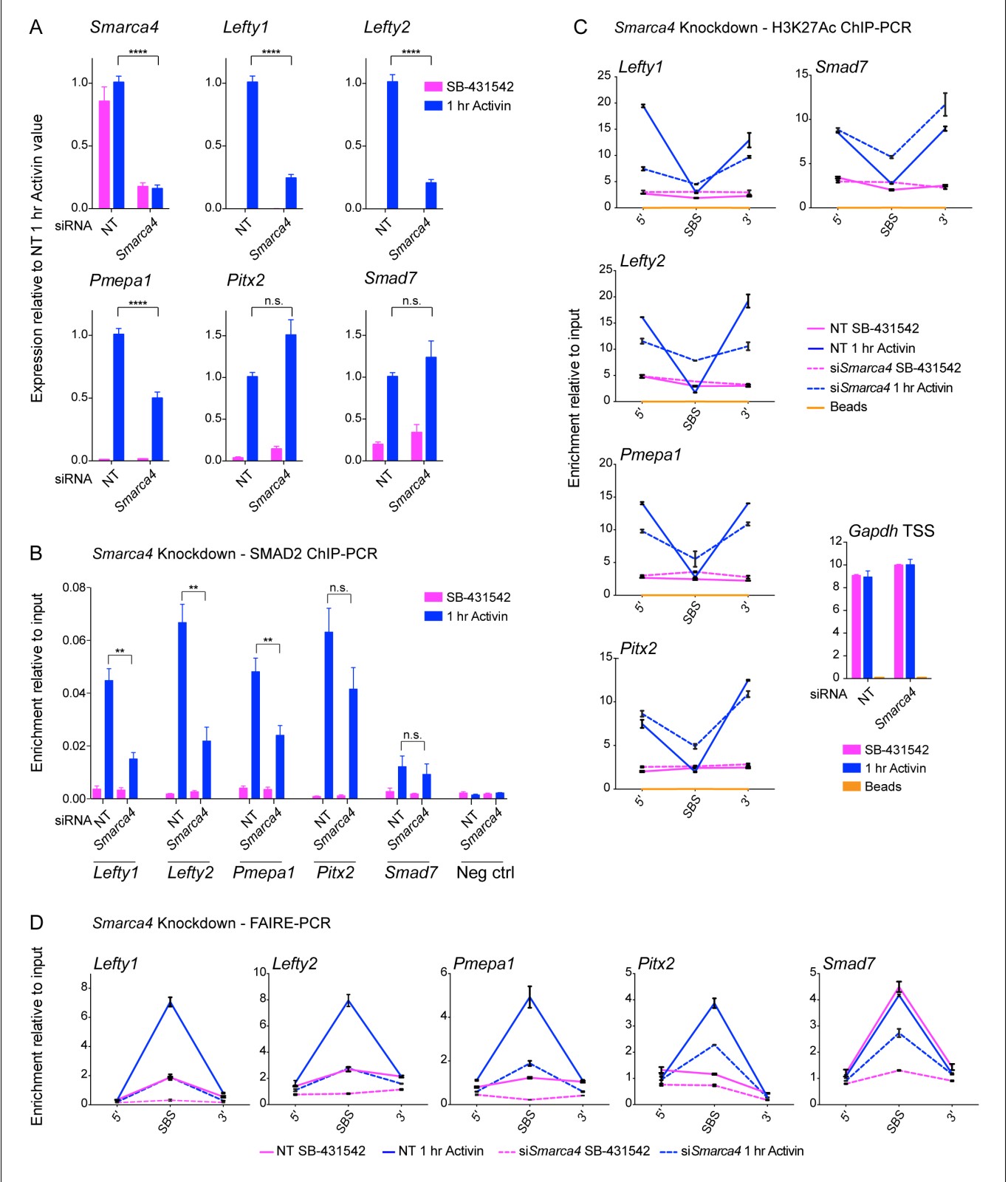

**Figure 8.** SMARCA4 is required for SMAD2 binding, nucleosome eviction and histone acetylation at a subset of Activin target genes. (**A**) P19 cells were transfected with either non-targeting (NT) or *Smarca4* siRNAs. Cells were then signal inhibited (SB-431542) or stimulated with Activin for 1 hr after SB-431542 washout. They were assayed for transcription by qPCR. The data shown are means ± SEM from four independent experiments. (**B–D**). Samples were prepared as for *A* and assayed for SMAD2 ChIP-PCR (**B**), H3K27Ac ChIP-PCR (**C**), or FAIRE-PCR (**D**) on the selected SBSs indicated. Plotted in B

*Figure 8 continued on next page*

*Figure 8 continued*

are the means and SEM of four independent experiments. Neg ctrl, negative control. The data in C and D are from a representative experiment of three (means ± SD). See **Figure 8—figure supplement 1** for the averages of the three experiments and the statistical analyses. In A and B, [****] corresponds to a p value of < 0.0001. [**] corresponds to a p value of < 0.01; n.s., not significant.

The following figure supplement is available for figure 8:

**Figure supplement 1.** The role of SMARCA4 in SMAD2-mediated chromatin remodeling.

already activated by SMAD2 and may reflect localized chromatin remodeling. In other cases, these delayed peaks occur at genes that are induced with delayed kinetics (for example *Eomes* and *Trh*) and may reflect newly synthesized co-factors recruiting SMAD2 to secondary target genes (**Figure 9**). There are many examples in the literature of SMAD complexes interacting with distinct TFs or switching their partner TFs during differentiation (**Brown et al., 2011**; **Faial et al., 2015**; **Mullen et al., 2011**).

Our data further suggest that rather than transcription termination resulting from loss of SMAD complexes from enhancers, SMAD2 must recruit secondary repressors to either attenuate transcription (induced sustained genes), or return it to basal levels (transiently induced genes). This mechanism also seems to be true of some of the repressed genes, which are initially transiently activated in response to Activin, prior to their repression. We have not yet provided direct evidence for this secondary repression mechanism, but possible candidates for such repressors are TGIF, SKI, SKIL, ZEB1/2 or EVI-1 that can associate with activated SMAD2/3 to repress transcription (**Deheuninck and Luo, 2009**; **Kurokawa et al., 1998**; **Postigo et al., 2003**; **Tsuneyoshi et al., 2012**; **Wotton et al., 1999**). Thus, we conclude that a transcriptional network is established downstream of Activin stimulation that modulates the transcriptional response at later time points.

During the course of this study we have compared the influence of ligand dose versus duration of signaling on the transcriptional response to NODAL/Activin signaling. Our data suggest that although some genes show a dose-dependent response, the major determinant of transcriptional output is signaling duration for the reasons outlined above. These results fit well with the observation that it is duration of NODAL signaling in zebrafish embryos, rather than amplitude, that determines cell fate specification (**Hagos and Dougan, 2007**; **Schier, 2009**).

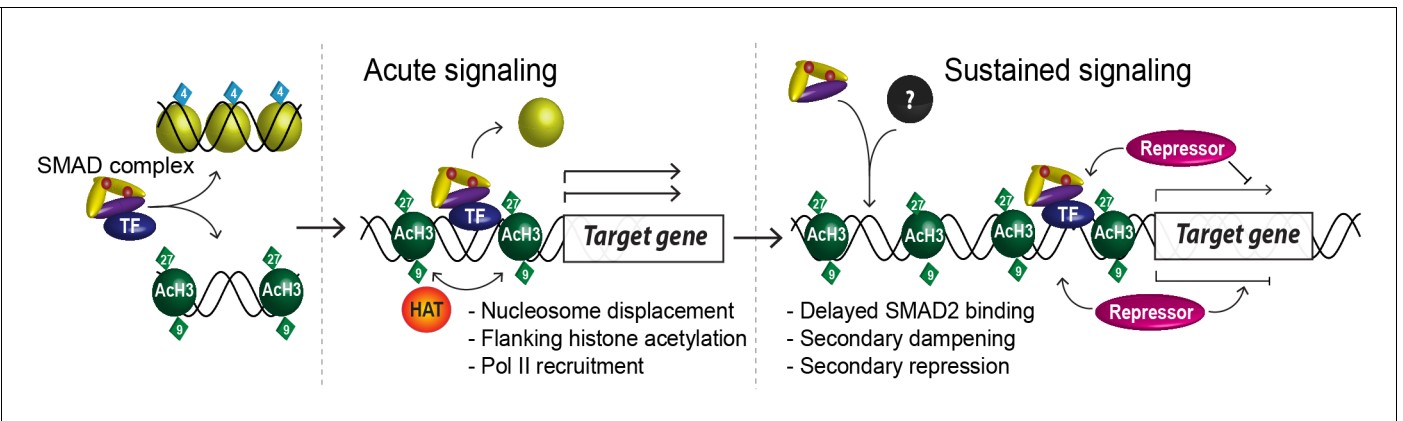

**Figure 9.** A model of dynamic SMAD2-dependent transcription. The two modes of SMAD2 binding to either acetylated (green diamonds) nucleosome-depleted chromatin or closed, non-acetylated chromatin marked by H3K4Me1 (blue diamonds) upon Activin stimulation from the SB-431542 state are depicted. SMAD2 binds in conjunction with FOXH1 at some targets or with distinct TFs at others. Once bound, SMAD2-containing complexes locally increase H3K27Ac and H3K9Ac via recruitment of HATs, and can induce nucleosome displacement, which for 'baseline off' genes requires SMARCA4. Upon sustained signaling, SMAD2 may be recruited to new targets in a delayed manner or already-bound SMAD2 may recruit repressors to dampen or inhibit transcription at later time points.

## SMAD2-induced chromatin remodeling

Tracking chromatin landscape changes over time at genes induced or repressed with different kinetics in response to Activin led us to the striking finding that at some loci SMAD2 binds to inactive chromatin. Its binding results in rapid nucleosome displacement and histone H3 acetylation on either side of the SMAD2-binding site. This subsequently leads to Pol II recruitment and acetylation at the TSS of the target gene with kinetics following the expression pattern of the transcript. The ability of activated SMAD2 to bind inactive chromatin and remodel it was unexpected, as it was previously suggested that SMAD2/3 only binds sites already occupied by master TFs, and that NODAL/Activin signaling does not affect nucleosome positioning at POU5F1-shared binding sites in ESCs (*Mullen et al., 2011*). We uncovered this alternative mode of SMAD2 binding because we investigated a timecourse of signaling from a signal-inhibited baseline, as opposed to focusing on constitutive signaling states.

Nucleosome remodelers have been implicated in SMAD2-mediated transcription, with the SWI/SNF ATPase SMARCA4 being required for a subset of TGF-$\beta$ responses in both P19 and human epithelial cells (*Ross et al., 2006*; *Xi et al., 2008*). BPTF is an ISWI nucleosome remodeler that also interacts with SMAD2/3 and mediates NODAL/Activin-induced visceral endoderm differentiation in early mouse embryos (*Landry et al., 2008*). However, until now it was unclear whether nucleosome remodeling was a prerequisite for SMAD2-containing complexes to find their target sites or a consequence of SMAD2-mediated transcription. We have shown that for 'baseline off' genes SMARCA4 is required for both SMAD2 binding and subsequent chromatin remodeling and nucleosome eviction. Interestingly SMARCA4 was not required at the *Pitx2* SBS, even though it exhibits some of the same characteristics as the *Lefty1* and *Lefty2* SBSs (FOXH1-dependent and closed unacetylated chromatin at the SBS). This suggests that the chromatin remodelers involved in SMAD2 binding and nucleosome eviction are gene specific, and that additional factors are involved in their recruitment.

Importantly, we found no evidence for a pioneer factor stably bound prior to SMAD2 recruitment at silent chromatin. In this respect these SBSs are analogous to so-called latent enhancers that are bound by stimulus-activated and lineage-determining TFs upon macrophage stimulation (*Ostuni et al., 2013*), and we will refer to them henceforth as such. Our data also suggest that the latent enhancers in P19 cells may be marked by H3K4 monomethylation, known to mark active and poised enhancers (*Zentner et al., 2011*). We demonstrated its presence at the *Lefty1* enhancer both in the SB-431542 state and after acute and chronic Activin induction. It remains unclear, however, how this modification is established prior to TF binding.

SMAD2 binding correlates with an acute induction of H3K27Ac and H3K9Ac either side of the SBS, suggesting that SMAD2 induces acetylation through recruitment of HATs. H3 acetylation is limited to one to two nucleosomes flanking the SMAD2 peak. SMADs have been shown to bind the HATs EP300, PCAF and GCN5 (*Ross and Hill, 2008*). We have demonstrated that EP300 is enriched together with SMAD2 at enhancer loci, and intriguingly, we previously showed that EP300 recruited to a chromatin template in vitro by DNA-tethered SMAD2 preferentially acetylates H3 over H4 (*Ross et al., 2006*). Overall we have observed that in response to acute Activin signaling, H3K9Ac and H3K27Ac are induced similarly. Although H3K27Ac is predominantly catalyzed by EP300 (*Calo and Wysocka, 2013*), H3K9Ac is thought to be catalyzed by GCN5/PCAF or TIP60 (*Karmodiya et al., 2012*). Thus activated SMAD2 likely recruits several distinct HAT complexes to mediate localized H3 acetylation.

## The role of SMAD2-cooperating transcription factors

Using motif enrichment analysis and siRNA-mediated knockdown we have demonstrated that a subset of NODAL/Activin targets relies on FOXH1 and/or POU5F1. For a selection of latent enhancers (for example *Lefty1, Lefty2,* and *Pitx2*) FOXH1 recruits the active SMAD2-containing complex. FOXH1 is a member of the Forkhead family of TFs and one of the best characterized SMAD2 binding partners (*Chen et al., 1996*; *Randall et al., 2002*; *Silvestri et al., 2008*). We had hypothesized that FOXH1 would act as a pioneer factor. However, we found that FOXH1–SMAD2 complexes bind their target sites only in response to signaling, implying that they bind cooperatively. This mode of binding and the requirement for chromatin remodeling complexes suggests that SMAD2 may bind chromatin through what has recently been termed the dynamic assisted loading mechanism (*Swinstead et al., 2016*). At the latent enhancers, FOXH1 is required for Activin-induced SMAD2

recruitment, and hence for ligand-induced nucleosome displacement and histone acetylation. Importantly, FOXH1 is not the only SMAD2-recruiting TF that can bind to latent enhancers. Activin-induced accessibility of the upstream enhancer of the *Pmepa1* gene does not require FOXH1 and must be dependent on an as yet unidentified TF.

In addition to the latent enhancers, we found many examples of enhancers that were already nucleosome depleted and flanked by acetylated nucleosomes, such as the SBSs flanking *Pou5f1* and *Trh*. In these cases, the genes exhibit a baseline of expression, indicating that a TF is already bound. POU5F1 itself is a good candidate for creating a favorable environment at these enhancers for SMAD2 to bind and potentiate transcription. Indeed, the fact that the POU5F1 canonical binding site is not centrally enriched under SMAD2 peaks argues for an indirect role for POU5F1 in SMAD2 recruitment.

### Concluding remarks

In conclusion, our study defines the sequence of events elicited directly by SMAD2 that occur on chromatin in response to signal activation. Our newly established paradigms highlight that there is no single generic mechanism of SMAD2-mediated transcription, but rather it is gene specific and can occur in an acute or delayed fashion and can be activatory or repressive. The initial transcriptional profile is modulated at later time points as a result of both delayed SMAD2 binding, and subsequent binding of repressors to already occupied SBSs. Future work will aim to identify TFs and enzymes that collaborate with the SMADS to modulate the transcriptional response over time, allowing cells to correctly execute gene expression programs in response to TGF-$\beta$ superfamily signaling.

## Materials and methods

### Cell line origin and authentication

P19 cells (*Rudnicki and Mcburney, 1987*) were obtained from Grace Gill (Harvard Medical School), C2C12 cells (*Bains et al., 1984*) were obtained from Richard Treisman (Francis Crick Institute) and EpH4 cells (*Reichmann et al., 1989*) were obtained from Harmut Beug (IMP, Vienna). All three cell lines were banked by the Francis Crick Institute Cell Services, were certified negative for mycoplasma and validated as of mouse origin. In each case their identity was authenticated by confirming that their responses to ligands and their phenotype was consistent with published history. All three cell lines were cultured in Dulbecco's Modified Eagle Medium containing 10% FCS.

### Cell treatments and siRNA transfections

P19 cells stably expressing MYC-FOXH1 were generated via transfection with a plasmid encoding MYC-tagged FOXH1 (*Labbé et al., 1998*) and pSUPER-retro-puro (*Grönroos et al., 2012*), and selecting for puromycin-resistant stable clones. NODAL/Activin signaling was inhibited by overnight incubation with 10 μM SB-431542 (Tocris Bioscience, UK), which was washed out three times with PBS prior to stimulation with 20 ng/ml Activin (Peprotech, New Jersey, USA) in full media for different times. For the SB-431542 condition, after washout, cells were incubated for 1 hr in 10 μM SB-431542 in full media, providing a control for a possible transient effect of serum stimulation in the 1 hr Activin samples. The untreated condition represents a chronic signaling state as a result of autocrine production of NODAL and GDF3. For the SB-431542 chase experiment, cells were treated as described above in the control samples, or 10 μM SB-431542 were added to the cells after 1 hr of Activin treatment for different times. Cycloheximide and emetine (Sigma, UK) were used at 5 μg/ml and 10 μM respectively. Actinomycin D (Sigma) was used at 6 μM and curve fitting of the experimental data was performed using the one phase decay function of Prism 6 software, with least squares fit as the fitting method. siRNA transfections were carried out using Lipofectamine RNAiMAX (Thermo Fisher Scientific, Massachusetts, USA) for 72 hr (48 hr for *Pou5f1*) at a final concentration of 20 nM. siRNAs were from Dharmacon (Colorado, USA) and are listed in *Supplementary file 3*.

### RNA and protein biochemistry

Total RNA was extracted using Trizol (Thermo Fisher Scientific) according to the manufacturer's instructions. cDNA synthesis and qPCRs were performed as described (*Grönroos et al., 2012*). Primer sequences are listed in *Supplementary file 3*. Total cell lysates were generated as described

(*Germain et al., 2000*), and Western blotted using the antibodies listed in *Supplementary file 3*. Western blots were imaged and quantified using an ImageQuant 4000 mini and ImageQuant TL 8.1 software (GE Healthcare, UK).

## Generation of FOXH1 antibody

A peptide corresponding to amino acids 16–31 of mouse FOXH1 was used to generate rabbit anti-sera against FOXH1. The antibody was affinity purified using a column of SulfoLink™ resin coupled to the peptide according to the manufacturer's instructions (Thermo Scientific, cat # 44999). We validated the antibody in several different ways. We proved that it recognised Myc-FOXH1 in a stably overexpressing P19 cell line by Western blot (*Figure 7—figure supplement 3*). Moreover, we compared its activity in a ChIP-PCR on wild-type P19 cells with that of a Myc antibody in the P19 cell line that expresses Myc-FOXH1 at approximately endogenous levels and found the results to be equivalent.

## Deletion of *Lefty1* and *Lefty2* SBSs by CRISPR/Cas9

To avoid problems of heterogeneity in the P19 population, we first isolated a clone of P19 cells that had the same characteristics as the P19 pool with respect to gene expression profiles and pSMAD2 induction patterns in response to Activin treatment. DNA oligos corresponding to the sgRNA sequences (see *Supplementary file 3*) were cloned into pSpCas9(BB)−2A-GFP (PX458) (*Ran et al., 2013*), and the plasmids were transfected into P19 cells using Lipofectamine 2000 (Invitrogen, California, USA) according to the manufacturer's instructions. 48 hr after transfection, the GFP-positive cells were FACS-sorted into 96 well plates. Resulting single cell clones were screened by PCR using primers flanking the guide sites (*Supplementary file 3*). The PCR fragments of positive clones were sequenced to identify the CRISPR/Cas9-mediated deletions.

## Chromatin immunoprecipitation (ChIP)

Cells were crosslinked using 1% formaldehyde for 10 min at room temperature, followed by lysis in 5 mM HEPES (pH 8), 85 mM KCl, 0.5% NP-40. After centrifugation, nuclei were washed in 5 mM HEPES (pH 8), 85 mM KCl and then sonicated to 100–400 bp using 8–12 30" on-off cycles in 50 mM Tris-HCl (pH 8), 10 mM EDTA, 1% SDS using a Bioruptor Sonicator (Diagenode, Belgium). The resulting chromatin was diluted 5x, pre-cleared with protein A/G Dynabeads (Invitrogen) and then incubated overnight with antibodies in immunoprecipitation (IP) buffer (50 mM HEPES (pH 7.5), 1% Triton X-100, 150 mM NaCl). Protein A or G Dynabeads (100 μl) blocked with BSA were then added for 6 hr, followed by six washes in IP buffer containing 0.1% sodium deoxycholate, 0.1% SDS and 1 mM EDTA, with the last three additionally containing 500 mM NaCl. A wash was then performed with 10 mM Tris-HCl pH 8, 250 mM LiCl, 1 mM EDTA, 1% NP-40 and 0.5% sodium deoxycholate followed by a final wash in 10 mM Tris-HCl pH 8, 1 mM EDTA. Chromatin was eluted in 1% SDS, 0.1 M NaHCO$_3$. De-crosslinking and Proteinase K digestion was carried out at 65°C overnight. The resulting enriched chromatin was cleaned up using a QiaQuick PCR purification kit (Qiagen, Germany) according to the manufacturer's instructions. qPCR was carried out as for gene expression analysis, with a standard curve consisting of 6 points and three-fold dilutions. Values for each IP sample were normalized relative to corresponding input chromatin for the same treatment. Primer sequences and antibodies are listed in *Supplementary file 3*.

## FAIRE-PCR

Samples for FAIRE-PCR were crosslinked, lysed and sonicated as for ChIP, followed by FAIRE as described (*Simon et al., 2012*).

## RNA-sequencing

All RNA-seq experiments were performed as biological duplicates. Where the RNA-seq was performed in the presence of protein synthesis inhibitors, cycloheximide or emetine were added after the PBS washout, 5 min before the addition of Activin or SB-431542. Total RNA was prepared using Trizol followed by a clean up with an RNeasy kit (Qiagen). The quality of the RNA was assessed using a Bioanalyzer (Agilent, California, USA). Libraries were prepared using the TruSeq Stranded mRNA

Sample Prep Kit (Illumina, California, USA). 101 bp single end reads were generated using an Illumina HiSeq 2500.

Read quality was assessed with FastQC (*Patel and Jain, 2012*). Quality score distributions across the reads were within acceptable limits with the majority of the reads having a Phred score above 30. There was no observable position-specific base bias within the reads. Read duplication levels were within the limits of what would be expected for RNA-seq, with the majority of reads being unique. There were no over-represented sequences. The 101 bp single-end read data were aligned to the mm10 version of the mouse genome using TopHat2 (*Kim et al., 2013*). A transcriptome guide file derived from the knownGene annotation table available from the University of California, Santa Cruz (UCSC) Genome Bioinformatics site was provided to focus alignments towards characterized transcripts. Non-unique alignments were removed prior to further analysis. The data were quantified at the gene level by counting reads aligning within each gene's exon structure. Reads partially aligning outside exon boundaries were excluded from the quantification using the summarizeOverlaps function from the GenomicAlignments Bioconductor library with mode='IntersectionStrict' (*Huber et al., 2015*). The mean RNA-seq gene alignment rate was 26.5 million reads.

Genes with low read counts are susceptible to sampling error and as a consequence their counts can be unreliable. We therefore removed genes with a mean count across all samples below the 40th percentile of counts across all genes and all samples. This effectively removes the non-expressed and lowly expressed genes prior to statistical analysis. We accounted for differences in total read counts observed between the different samples using the normalization strategy found in the DESeq package from Bioconductor (*Anders and Huber, 2010*). This approach is equivalent to scaling to the total number of reads, however it is biased towards non-differential genes. This approach helps to negate instances where a small number of highly expressed and differential genes can have a strong effect on the total read count.

To determine differentially-expressed genes we used a negative binomial distribution implemented in DESeq to model the data. We ran pairwise comparisons of each time point against the SB-431542 sample to determine time point-specific differential expression, and identified differentially-expressed genes using an FDR threshold of 0.05 and a $\log_2$FC filter of 0.7. We further manually filtered this list by discarding genes with <30 reads across all treatment conditions, as these likely represent background noise. We also removed genes only showing a significant change in the 'Untreated' condition, which might be due to long-term serum depletion of that sample compared to the others. This gave a total of 747 differentially-regulated genes.

## Further clustering of target genes

To cluster the genes according to their kinetic expression patterns, the following rules were used.

Induced sustained: 1 hr Activin $\log_2$FC $\geq$0.7; 8 hr Activin $\log_2$FC >1 hr Activin $\log_2$FC; Untreated $\log_2$FC $\geq$0.

Transient induced: 1 hr Activin $\log_2$FC $\geq$0.7; 8 hr Activin $\log_2$FC <1 hr Activin $\log_2$FC and $\geq$0; Untreated $\log_2$FC $\leq$0.7 and $\geq$−0.7

Delayed: 1 hr Activin $\log_2$FC $\leq$0.7; 8 hr Activin $\log_2$FC $\geq$0.7; Untreated $\log_2$FC >0.

Repressed: 8 hr Activin $\log_2$FC $\leq$0.7; Untreated $\log_2$FC $\leq$0.

In all cases, the $\log_2$FC was relative to the SB-431542 sample. For classification between 'baseline on' and 'baseline off', a gene was designated as 'baseline off' if less than 30 reads were detected in the SB-431542 sample. For classification as direct or indirect, a gene was defined as indirect if in both the cycloheximide- and emetine-treated 8 hr sample the $\log_2$FC did not reach >0.5 (for induced sustained and delayed genes) or < −0.5 (for repressed genes), or if 8 hr Activin $\log_2$FC >1 hr Activin $\log_2$FC (for transient induced genes).

## ChIP-sequencing

Samples for ChIP-seq with the corresponding inputs were prepared as described above for ChIP-PCR. Following end repair, poly-A-tailing and adapter ligation, Illumina TruSeq ChIP sample preparation kits were used to generate the libraries. The Illumina kit Phusion enzyme was replaced by Kapa HiFi HotStart ready mix (Kapa Biosystems, Cape Town, South Africa). The PCR was run before gel isolation using the Invitrogen SizeSelect E-gel system (SizeSelect gel protocol, Thermo Fisher Scientific). Post PCR we used AMPure XP beads (AMPure bead protocol, Beckman Coulter, Inc.) at a 1:1

ratio to maintain size integrity. Samples were multiplexed and 51 bp single end reads were generated on an Illumina HiSeq 2500. The raw reads were aligned to the mouse mm10 genome assembly using BWA 0.6.2-r126 (*Li and Durbin, 2009*), with a maximum mismatch threshold of 2 permissible within a seed length of 51 bp. All other parameters were kept as default. For the Pol II and histone ChIP-seq datasets the alignments were post-processed with picard-tools 1.107 for the removal of reads that could have arisen from PCR duplication (http://sourceforge.net/projects/picard/). All ChIP-seq experiments were performed as biological duplicates.

## SMAD2 peak calling, annotation and generation of a SMAD2 consensus peak list

To identify regions enriched with reads when compared to an input control we used MACS (*Zhang et al., 2008*) (q-value 0.05). Peaks were annotated in terms of distance to nearest gene (user-defined) using ChIPPeakAnno (Bioconductor) (*Zhu et al., 2010*). We associated SMAD2-binding loci to the closest differential gene as determined by our RNA-seq and ChIP-seq for Pol II Ser5 data (max distance 100 kb from TSS/TTS). To quantify our SMAD2 binding sites we scaled the total number of mapped reads per sample to $40 \times 10^6$ and counted reads mapping to SMAD2 binding loci.

To quantify gene-associated SMAD2 activity across the four time points we constructed a set of consensus binding sites by collapsing all binding sites identified from all comparisons. Binding loci were merged if they shared an overlap of at least a single nucleotide. Note that in our dataset the mean length of overlap between two MACS peaks was 200 bp, showing that the majority of the peaks have a 'sizeable' overlap. We quantified gene-associated SMAD2 activity by summing the normalized read counts across consensus loci that were associated with an individual regulated gene (SMAD2 footprint). Duplicate reads were removed prior to consensus peak quantification and reads were shifted to account for the fragment effect present in sequencing data.

To select a list of SMAD2 peaks to take forward, we filtered out those peaks with low MACS fold enrichment scores (<3.0). Furthermore, we determined whether a peak contained sequences derived from repetitive elements by determining overlap with repeats reported in the repeatmasker track, UCSC Genome Browser. Peaks mapping to highly conserved repetitive sequence were removed.

## Pol II (Ser5P and Ser2P) and H3K27Ac/H3K9Ac ChIP-seq analysis

Accounting for variations in the sampling of the underlying sequence library caused by differences in the total number of reads sequenced is not straightforward for ChIP samples. Using total mapped reads has been shown to be unreliable, since variations in enrichment efficiency can have dramatic effects on the proportion of reads attributed to signal. A more appropriate normalization strategy is to use only reads derived from regions with high signal and low variance across all samples. Therefore, we used DiffReps 1.55.4 (*Shen et al., 2013*) to calculate sample-specific normalization factors for the Pol II (either phosphorylated on Ser2 or Ser5 of the CTD), H3K27Ac and H3K9Ac ChIP-seq datasets. Furthermore, to be able to compare the signal for the same protein or modification from different treatments, the DiffReps normalization factors generated per time point were adjusted so that the geometric mean of the SB-431542 samples was constant across all comparisons from the same IP.

We also used DiffReps to determine sites of differential enrichment of Pol II (either phosphorylated on Ser2 or Ser5 of the CTD) at each time point relative to SB-431542. For each ChIP-seq experiment, replicate samples for each time point were compared to the SB-431542 sample. The corresponding input samples were also processed with DiffReps for background noise estimation. The '–nsd' and '–frag' parameters were set to 'broad' and '200', respectively, and all other parameters were kept as default. Sites of differential binding for Pol II Ser2P and Ser5P from all the comparisons were overlapped with respect to knownGene gene loci from the UCSC with the allowance for 2 kb of flanking sequence. More specifically, for each comparison, the sum of all differential bases found to overlap a given gene was calculated, and this was reported as a ratio of the length of the gene interval. Through inspection of the dataset it was found that genes exhibiting an overlap ratio ≥0.09 in any one of the comparisons were the most reliable. This gene list was then manually filtered to remove microRNAs and false positives, giving a dataset of 410 genes. This dataset represents the annotated genes which clearly show differential Pol II occupancy in response to signaling.

### Definition of the high confidence dataset of target genes and associated binding sites

Genes differentially expressed according to RNA-seq, as well as genes found to contain differential Pol II occupancy at any time point, were taken forward as SMAD2 target genes. Overlapping genes between these two lists were selected. Outliers on either side were additionally included based on the presence of SMAD2 peaks conserved between the two biological ChIP-seq replicates within 100 kb of an annotated TSS or TTS. Moreover, genes that had differential Pol II binding, but did not pass the stringent filter used for the RNA-seq data, were included if they were found to be differentially expressed according to less stringently filtered RNA-seq analysis and by manual inspection of the data. This led to a final list of 140 genes, associated with 478 SMAD2 consensus peaks.

### Assessing histone modification or Pol II enrichment around SMAD2 peaks and target gene TSSs using metaprofiles

Histone modification changes or Pol II enrichment across the time points for specific peak and gene sets were visualized using metaprofiles. In all cases the reads for the two biological replicates were added together to generate more read depth. The read depth vectors for loci across SMAD2 binding sites (the average MACS-called summit for each consensus peak) and SMAD2 target gene TSS loci were obtained. These vectors were then adjusted using the normalization factors generated by DiffReps described above, and metaprofiles were constructed by averaging per-nucleotide read depths across contributing loci. The metaprofiles were smoothened using local polynomials. In the case of TSS-centric profiles, genes in a reverse orientation were accounted for, and all plots run from 5' to 3'.

### Hierarchical clustering

To determine the similarity between the histone modifications at SMAD2 binding sites across the time points we clustered the DiffReps-normalized read count data. The normalized read count profiles were summed to give a single value per locus per sample (±2.5 kb relative to the TSS). We employed hierarchical clustering using euclidean distance to group the regions.

To quantify the $\log_2$FC of Pol II (either Ser5P or Ser2P), the total numbers of reads mapping to the entire genomic extent of a gene (±2 kb) was determined, adjusting the counts from each sample by its DiffReps normalization factor. The mean values of these counts across the replicates was taken and compared to the mean normalized replicate value of the relevant SB-431542 sample. Hierarchical clustering was employed using euclidean distance. A similar analysis was performed with the H3K9Ac and H3K27Ac datasets focusing on either the TSS ± 2.5 kb or the SMAD2 binding sites.

### Correlation of SMAD2 binding with overall levels of histone acetylation

From the analysis of histone modifications at SMAD2 binding sites described above, we defined regions of high or low acetylation for H3K9 and H3K27 by selecting the nodes obtained via hierarchical clustering that show either high or low H3K9Ac/H3K27Ac (see *Figure 6A*). Taking the overlap between the H3K9Ac and H3K27Ac datasets, we defined 72 'low acetylation' SMAD2 peaks and 44 'high acetylation' SMAD2 peaks. To define peaks with low or high acetylation in the SB-431542 sample, the 100 most or least enriched acetylated peak regions in this sample were selected for each modification and the intersection of peaks selected. This resulted in 60 'low H3Ac SB baseline' and 63 'high H3Ac SB baseline' peaks. To define the low acetylation peaks in the SB-431542 sample segregated according to whether they remained lowly acetylated upon Activin induction or were increased, we selected the 120 peaks for which acetylation is low for both modifications in the SB-431542 state. These were divided into 62 peaks that show a 'high increase' and 58 that show 'no increase' in response to signaling, using a $\log_2$FC of $\geq$0.7 cutoff for both acetylation states in the 8 hr Activin sample relative to the SB-431542 sample.

### Public availability of data

The ChIP-seq and RNA-seq data have been submitted to the NCBI Gene Expression Omnibus (GEO) under the accession number GSE77488.

## IGV browser displays

For visualization of ChIP-seq data using the IGV browser, Pol II and histone acetylation tracks are shown normalized using the DiffReps-generated factors, read extension by 100 bp and smoothing over 10 bp windows, while the SMAD2 tracks have been extended and smoothened. The H3 tracks represent the raw coverage obtained from the aligned reads.

## Gene ontology analysis

Metacore was used to identify enriched gene ontology signaling pathways, molecular function and biological processes within the SMAD2 target gene list. Default settings were used and also background lists provided by the program itself.

## Motif enrichment analysis

Taking all 478 consensus peaks associated with a SMAD2 target gene, the MACS-defined summits from each contributing MACS-called peak were selected (757 in total). The 500 bp sequence surrounding each summit was extracted and submitted for known and de novo motif discovery using the MEME-ChIP suite. Default settings were used with the exception of MEME (site distribution – any number of occurrences; total number – 20). To search for specific occurrence and location of user-defined motifs within this dataset, Fuzznuc was used (*Rice et al., 2000*). Motif strings were designed based on published binding sites found in the Genomatix, Uniprobe, Jaspar and Motifmap databases. Three background datasets were used for comparison. (1) SMAD2 peak sequences were shuffled in a tri-nucleotide manner. (2) Searches were carried out on 5000 enhancer peak summits (300 bp each) as obtained from the ENCODE database for MAFK, ZC3H11A, HCFC and ZNF384 in mESCs. (3) Sequences were extracted from the mouse genome (9993 sequences, 500 bp in length), matched for distance from any SMAD2-regulated target gene at 8 hr Activin signaling, but in this case using genes which did not change their expression in response to the pathway.

## Statistical analysis

Experiments were performed at least twice independently (biological duplicates) and the majority were performed at least three times. Within each qPCR experiment, technical duplicates were run. Statistical analyses were performed using an unpaired t-test unless otherwise specified. $p < 0.01$ was considered statistically significant.

# Acknowledgements

We thank Liliana Attisano for the Myc-Foxh1 plasmid, Thodoris Petrakis for generating the MYC-FOXH1 cell line and Marc-Sebastian Walter for help in generating the *Lefty1/2* SBS deletion lines. We are grateful to the Francis Crick Institute Genomics-Equipment Park and to members of the Advanced Sequencing Facility. We thank Stuart Horswell for advice and Francesco Gualdrini, Mike Howell and members of the Hill lab for comments on the manuscript. This work was supported by the Francis Crick Institute which receives its core funding from Cancer Research UK (FC001095), the UK Medical Research Council (FC001095), and the Wellcome Trust (FC001095). In addition, this work was supported by the European Commission Network of Excellence EpiGeneSys (HEALTH-F4-2010-257082). We confirm that none of the authors have any financial interest in this work.

# Additional information

## Funding

| Funder | Grant reference number | Author |
|---|---|---|
| Francis Crick Institute | FC001095 | Davide M Coda<br>Tessa Gaarenstroom<br>Philip East<br>Harshil Patel<br>Daniel S J Miller<br>Anna Lobley<br>Nik Matthews<br>Aengus Stewart |

| | | Caroline S Hill |
| --- | --- | --- |
| European Commission | HEALTH-F4-2010-257082 | Caroline S Hill |

The funders had no role in study design, data collection and interpretation, or the decision to submit the work for publication.

## Author contributions

DMC, TG, Conceptualization, Data curation, Formal analysis, Validation, Investigation, Methodology, Writing—original draft, Writing—review and editing; PE, HP, Data curation, Formal analysis, Visualization, Methodology, Writing—review and editing; DSJM, Data curation, Formal analysis, Investigation; AL, Data curation, Formal analysis; NM, Investigation; AS, Data curation, Formal analysis, Supervision; CSH, Conceptualization, Supervision, Funding acquisition, Writing—original draft, Project administration, Writing—review and editing

## Author ORCIDs

Caroline S Hill, http://orcid.org/0000-0002-8632-0480

# Additional files

## Supplementary files

• Supplementary file 1. RNA-seq and ChIP-seq datasets.

• Supplementary file 2. Gene set enrichment analysis

• Supplementary file 3. List of reagents used in the study.

## Major datasets

The following dataset was generated:

| Author(s) | Year | Dataset title | Dataset URL | Database, license, and accessibility information |
| --- | --- | --- | --- | --- |
| Gaarenstroom T, Coda DM, East P, Patel H, Miller DSJ, Lobley A, Matthews N, Stewart A, Hill CS | 2016 | Distinct modes of SMAD2 chromatin binding and remodeling shape the transcriptional response to Nodal/Activin signaling. | http://www.ncbi.nlm.nih.gov/geo/query/acc.cgi?acc=GSE77488 | GSE77488 |

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
