## [Decision Letter]

[Editors’ note: a previous version of this study was rejected after peer review, but the authors submitted for reconsideration. The first decision letter after peer review is shown below.]

Thank you for submitting your work entitled "Distinct modes of SMAD2 chromatin binding and remodeling shape the transcriptional response to NODAL/Activin signaling" for consideration by *eLife*. Your article has been evaluated by Jessica Tyler as the Senior Editor and three reviewers, one of whom is a member of our Board of Reviewing Editors. The reviewers have opted to remain anonymous.

Our decision has been reached after consultation between the reviewers. Based on these discussions and the individual reviews below, we regret to inform you that your work will not be considered further for publication in *eLife*.

This study was designed to examine the SMAD2 genomic response in P19 embryonic teratoma cells treated with activin. The study is thorough and includes a variety of genome-wide methods, including RNA-seq and ChIP-seq. Together, the experiments reported provide a very useful data resource. This represents a major strength of the manuscript. However, the broad scope of the data presented did not lead to sufficient focus on conclusions that provide new insight into mechanisms of Smad2 signaling (see appended reviews, below). The consensus opinion of the reviewers is that the manuscript needs to be re-organized to highlight data that supports new conclusions and that additional experimental analysis may be needed to fully establish such conclusions. This opinion led to the decision to return the manuscript to you at this time.

*Reviewer #1:*

This study was designed to examine the SMAD2 genomic response in P19 embryonic teratoma cells treated with activin. The study is thorough and includes a variety of genome-wide methods, including RNA-seq and ChIP-seq. Together, the experiments reported provide a very useful data resource. This represents a major strength of the manuscript.

Three major conclusions are reported by the authors:

1) Recruitment of SMAD2 to both open and closed regions of chromatin is detected. This is argued to be incompatible with a model of SMAD2 recruitment to pioneer transcription factors (PMID: 22036565 & PMID: 22036566), but for this function, pre-binding is not essential. Indeed, it has been previously reported that FOXH1 is recruited to chromatin following activin signaling (PMID: 21741376). Moreover, a mechanism for activin-stimulated recruitment of FoxH1 to closed chromatin has been reported to be mediated by TIF1g-Smad2/3 (PMID: 22196728). It is unclear that the present study extends these previous findings.

2) SMARCA4 contributes to SMAD2 function at some sites. As noted by the authors, this has been noted elsewhere (PMID: 16990801 & PMID: 18003620).

3) The authors report (subsection “SMAD2 induces changes in the chromatin landscape to regulate transcription”, second paragraph; data not shown) that no significant enrichments of H3K27Me3 or H3K9Me3 were detected near mad binding sites. This finding contrasts with PMID: 25805847 (for at least some activin-dependent genes), but there is no follow up on this observation.

On balance, this is a good study that provides an outstanding data resource that will be useful to researchers in the field of action/nodal signaling. However, enthusiasm for this study is dampened by the limited novelty of the major conclusions drawn by the authors.

*Reviewer #2:*

In this paper, the authors use several genome-wide approaches to investigate the mechanisms employed by Smad transcription factors to control transient gene expression profiles. A lot of data is generated and presented in this paper and from this the authors attempt to make several conclusions. However, overall, despite the large amount of work, the study lacks fundamentally new findings on how Smad transcription factors work. For example, the authors claim that Smad2 works through affecting polymerase recruitment and not pause release but the data are not entirely consistent with this view (see below). The authors also indicate that Smad binding kinetics do not necessarily correlate with gene expression profiles (and hence use different mechanisms at different genes). However, the authors do not take into account message stability, which is potentially reshaping the target gene expression profiles rather than a transcriptional effect per se. Another important issue is the inclusion of "media only" effects in timecourses of activin signalling and this is not really appropriate (see below) and impacts on the interpretation of the gene expression profiles. Two of the latter findings (i.e. involvement of FOXH1 and SMARCCA4) are interesting but are already well known and studied Smad effector proteins. Overall, the current paper is therefore not conclusive enough for publication, in part through the inclusion of too much information that is not always necessary. However, with streamlining and careful re-interpretation of the data, a more coherent story might emerge.

1) The inclusion of the "untreated" samples in parts B and C is not entirely valid. This is not the same as treating with activin for 48 hrs and is actually a different question being asked i.e. what is the difference between acute activin signalling responses and endogenously generated sustained signalling. This of course impacts greatly on the profiles in part C, where the last time point should be deleted. The "sustained" signal is not actually central to the story and could easily be deleted from the paper. Indeed, many of the later figures do not use this condition anyway.

2) Have the authors considered whether RNA stability is shaping the profiles in Figure 1? This could explain the difference between "induced sustained" versus "transient induced". If the authors look at the RNA polymerase profiles (as should report on transcription) across the same gene clusters, do they see the same effects?

3) In many places, the authors include "representative experiments" rather than biological replicates. As "representative" implies replicates have been done, then why are they not included? Can the authors explain their rationale for doing this? For timecourse experiments, this can be acceptable due to adjacent timepoints acting as replicates (e.g. Figure 2) but for other experiments this is less clear. For example, Figure 1—figure supplement 2 should contain averages of the replicate experiments as this is a simple knockdown experiment. The same applies elsewhere and should be justified.

4) It is not clear how the replicate ChIP-seq experiments were processed to get the final list of Smad targets. How reproducible were the two ChIP-seq experiments? Also in generating "gene associated Smad activity" it is not clear why the authors took a "1bp" overlap in peaks. If looking at the same peak, then this is not stringent enough as the summits could be up to 500 bp apart and a parameter like summit-summit distance or% peak overlap should be used instead. If attempting to just associated total Smad binding with a gene, then the peak overlaps are not relevant. What happens if alternative ways of merging the Smad signal are used?

5) Figure 3 is central to the authors' conclusions about whether pause-release is occurring or not. However, in the "baseline on" set, there is clear evidence of paused polymerase in the inhibited condition i.e. at the TSS. In contrast this is not the case in the "baseline off" samples. In the latter case, then "recruitment" must be the primary mechanism and in the former case, pause-release must be the initial event. After the initial events it is highly likely that both mechanisms are important in both cases (i.e. the polymerase must be recruited and move to the elongation phase at both types of site once the first polymerase has moved off). It is also apparent in Figure 3 that both Ser2 and Ser5 forms of the polymerase increase which again suggests both mechanisms are generally activated. Have the authors looked at total RNA polymerase levels by ChIP rather than the CTD marks which inevitably pick out the promoter-proximal and elongating polymerases.

6) Another issue arises from Figure 3 as it is clear that the "repressed" genes are actually transiently activated which is actually apparent from the RNAseq data. Interestingly these genes show a lot of evidence for paused polymerase at the TSS. Perhaps more importantly, looking back on Figure 1, all of the genes show increased activation at the 1 hr time point. This suggests a unifying mechanism of Smad action through upregulating the initial transcription levels. The authors invoke a mechanism of Smads recruiting additional co-repressors to then shape subsequent responses and there is no evidence presented to support this claim (and evidence should be provided for this claim to be made).

7) The authors claim that Smad has two modes of binding. One to open chromatin and one to closed chromatin. FAIRE data is provided to support this claim in a locus-specific manner (did the authors look genome-wide and what was the result?). Also, the authors plot H3 density across Smad binding regions. However, in Figure 5 there is very little change in H3 occupancy around the centre of the Smad binding region. Even in the "baseline off samples" (<30 regions), then the chromatin is still open at the Smad binding regions. Therefore, it appears that there is a window of accessibility already present to which Smad2 binds, and subsequently widens. Thus, the idea that Smad2 is binding differently is not really supported by the data but is in fact accessing a "nucleosome depleted" DNA region in all cases.

8) Related to point 7, the authors indicated that FOXH1 is not "pioneering" as binds at the same time as Smad2. However, if they are binding in the same complex, could FOXH1 still not be "pioneering in that context" i.e. the one that would access the closed chromatin?

9) In the text, it is stated that H3K9Ac and K27ac show the same patterns. However, they are clearly different in Figure 5. i.e. H3K9Ac is transiently induced at 1hr at all sites whereas H3K27Ac is sustainably induced. The former correlates with initial Smad binding and the initial induction of gene expression and suggests could be the critical event controlled by Smad2. It is not clear why the authors have not commented more on this or developed this line of investigation.

10) It was not clear whether the selected genes in Figure 7 and Figure 8 are in the "baseline off" category as they should be if testing chromatin accessibility issues.

11) It is essential that biological replicates are included in Figure 8. At the moment for example it is difficult to conclude much from Figure 8, as SMad2 enrichment goes down irrespective of the gene (i.e. not as stated in the text). Statistical significance (or not) would help clarify this. The effect of SMARCA4 is not clearcut. In part D for example, the FAIRE signal changes at Pitx2 and Smad7 show similar changes despite different baselines. The interpretation of this data could be that SMARCA4 is important for maintaining open chromatin whether inducible by Smad2 or not (i.e. important for all open chromatin).

*Reviewer #3:*

In the present study, the authors examined the role of Smad2 in activin/nodal signaling-induced gene expression in P19 embryonic teratoma cells. I understand that the authors made great efforts to finish this paper. They provide many interesting observations including multiple modes of Smad2 binding to chromatin and the involvement of SMARCA4 in expression of a subset of Smad2 target genes. I think that the paper significantly contributes to further understanding of the Smad signaling pathway. However, I have a few concerns as follows.

1) Activin/nodal signaling induces phosphorylation of Smad2 and Smad3 (as described in the third paragraph of the Introduction of this paper). However, the authors analyzed binding of Smad2 to chromatin but not that of Smad3. Is it appropriate to discuss activin/nodal signaling only referring Smad2? They should explain in the main text why they omitted arguments about Smad3. Other ChIP-seq papers on activin/nodal signaling analyzed both Smad2 and Smad3.

Yoon et al. Genes Dev. 25, 1654-1661, 2011 (not cited)

Kim et al. Dev. Biol. 357, 492-504, 2011 (cited)

Chiu et al. Development 141, 4537-4547, 2014 (not cited)

2) To conclude that Smad2 regulates target gene expression through de novo recruitment of pol II (subsection “SMAD2 regulates Pol II via recruitment”, last sentence), the authors should directly show that knockdown of Smad2 results in failure of pol II to be enriched around the TSSs of target genes. I am afraid that Smad3 may also be involved. Same to the description in the fourth paragraph of the subsection “SMAD2 induces changes in the chromatin landscape to regulate transcription”.

3) How did the authors validate specificity of the anti-FoxH1 antibody prepared by themselves?

---

## [Author Response]

[Editors’ note: the author responses to the first round of peer review follow.]

*[…] Reviewer #1:*

*This study was designed to examine the SMAD2 genomic response in P19 embryonic teratoma cells treated with activin. The study is thorough and includes a variety of genome-wide methods, including RNA-seq and ChIP-seq. Together, the experiments reported provide a very useful data resource. This represents a major strength of the manuscript.*

We were pleased to see that the reviewer recognized a strength of the paper as an outstanding data resource. We have now performed a large number of new experiments and analyses to corroborate the data and strengthen our new mechanistic insights into SMAD2 signalling. We have also rewritten the manuscript to better communicate the novelty of the major conclusions.

The point of the study, which I think didn’t come over sufficiently strongly in the original version, was to answer fundamental questions about how transcription factor binding regulates transcription. Over the last decade a huge effort has been put into genome-wide mapping of transcription factor binding sites, histone modifications and chromatin accessibility. However, it is still unclear whether transcription factor occupancy induces histone modifications, or if the chromatin landscape dictates recruitment. Furthermore, much of this mapping has been carried out in steady state conditions. As a result, we know little about the sequence of events that occur from transcription factor binding to RNA Polymerase II regulation, or about how transcription factor binding influences the temporal patterns of gene expression in the longer term. The best way to study these critical issues is to use a system where the transcription factor binds in response to extracellular signalling, and to do this we have chosen to focus on Activin/NODAL signalling.

In this paper for the first time we have delineated the sequence of events that occur from SMAD2 binding to transcriptional activation, and the mechanisms underlying them. Crucially, our work establishes new paradigms for signal-dependent transcriptional regulation.

1) We have provided unequivocal evidence that transcription factor binding causes histone modification and chromatin remodelling with the discovery that SMAD2 binds to closed chromatin and remodels it.

This conclusion has now been strengthened with inclusion of additional metaprofiles (revised Figure 5) and additional ChIP-PCRs showing unequivocally that binding of SMAD2 to the *Lefty1* and *Pmepa1* SBSs results in nucleosome eviction (revised Figure 5—figure supplement 1).

This discovery also overturns the dogma in the field resulting from the Young and Zon papers (PMIDs 22036565 and 22036566, see below) that SMADs act passively and only bind to ‘preconditioned’ chromatin.

2) We demonstrate that SMAD2 regulates RNA Polymerase II via de novorecruitment to target promoters. We have considerably strengthened this conclusion in the revised manuscript by analysing metaprofiles of Pol II Ser2P (see revised Figure 3).

3) We demonstrate that SMAD2 chromatin binding does not directly correlate with transcription of target genes, and thus have overturned the assumption in the field that transcription factor binding linearly equates with transcriptional kinetics.

We have now explored this latter issue in much more detail with an SB-431542 chase experiment (see Figure 1 and Figure 1—figure supplement 4). The results indicate that modulation of long-term responses to NODAL/Activin signalling requires on-going signalling. This provides a functional explanation for our observation that SMAD2 remains bound at regulatory regions of genes as their transcriptional profiles are modulated over time. It also supplies the previously missing crucial evidence that the underlying mechanism is via SMAD2 sequentially recruiting multiple regulators to chromatin.

*Three major conclusions are reported by the authors:*

*1) Recruitment of SMAD2 to both open and closed regions of chromatin is detected. This is argued to be incompatible with a model of SMAD2 recruitment to pioneer transcription factors (PMID: 22036565 & PMID: 22036566), but for this function, pre-binding is not essential. Indeed, it has been previously reported that FOXH1 is recruited to chromatin following activin signaling (PMID: 21741376). Moreover, a mechanism for activin-stimulated recruitment of FoxH1 to closed chromatin has been reported to be mediated by TIF1g-Smad2/3 (PMID: 22196728). It is unclear that the present study extends these previous findings.*

The reviewer argues that the two papers from the Young and Zon labs (PMID: 22036565 & PMID: 22036566), which conclude that master transcription factors (TFs) dictate cell type-specific responses to TGF-β/BMP signalling, could in fact be consistent with our finding that SMAD2 exhibits the two distinct modes of chromatin binding.

It is true that in principle, the activated SMADs could bind DNA simultaneously with the master TFs, but neither of these papers actually addressed this issue. They did not investigate whether the co-occupancy of SMADs with master TFs was a result of pre- binding of the master TF, or whether the SMADs and master TFs bind simultaneously. To do this they would have had to investigate the chromatin state and presence/absence of master TFs in the non-signalling state, which they did not. Our paper is the first to address this issue.

Moreover, the Young paper (PMID 22036565) favours the pre-binding model as it states that master transcription factors help direct SMAD3 binding by establishing open chromatin that contains SBEs, allowing SMAD3 to bind DNA and form a physical complex with the master transcription factors. As a result of these papers the dogma developed in the field that SMADs act passively and only bind to already accessible chromatin. Our paper overturns this dogma and is the first to show that the SMADs bind closed unacetylated chromatin and displace nucleosomes.

The reviewer mentions that it has previously been reported that FOXH1 is recruited to chromatin following Activin signalling (PMID: 21741376). These authors did report inducible binding of FOXH1 and SMAD2/3 at the *EOMES* and *GSC* genes in human ESCs. However, the predominant increase in FOXH1 binding that correlates with SMAD2/3 binding actually occurs at 24 hr post Activin stimulation. Therefore, it is not clear how this relates to the rapid simultaneous binding of FOXH1 and SMAD2 at closed chromatin that we observe. Moreover, they interpret the role of FOXH1 as a pioneer factor (see Discussion of Kim et al), which is not what we conclude.

The reviewer also mentions a paper (PMID: 22196728) which they say reports a mechanism for Activin-stimulated recruitment of FOXH1 to closed chromatin mediated by TIFγ-SMAD2/3. However, this paper does not address FOXH1 binding at all, and even for the SMAD2/3–SMAD4 complex binding to the proximal promoter element, it is not shown how binding of TIF1γ together with SMAD2/3 at the upstream site actually facilitates this.

*2) SMARCA4 contributes to SMAD2 function at some sites. As noted by the authors, this has been noted elsewhere (PMID: 16990801 & PMID: 18003620).*

The reviewer is correct that the interaction of SMAD2 with SMARCA4 and a functional effect on transcription has been noted previously. In fact, one of these papers came from my lab. It is not known, however, at what level SMARCA4 is required. Here we show definitely that it is required for the nucleosome displacement at SMAD2 binding sites at ‘baseline off’ target genes.

*3) The authors report (subsection “SMAD2 induces changes in the chromatin landscape to regulate transcription”, second paragraph; data not shown) that no significant enrichments of H3K27Me3 or H3K9Me3 were detected near mad binding sites. This finding contrasts with PMID: 25805847 (for at least some activin-dependent genes), but there is no follow up on this observation.*

We found no enrichment of H3K27Me3 or H3K9Me3 at Activin target genes in P19 cells. Because of this we did not investigate these marks further. Their absence was not due to a failure of these ChIPs though, as we did find other regions of the genome enriched in these marks, such as the *HOX* clusters. The paper that the reviewer refers to uses hESCs and is focused on H3K4me3, which we did not investigate. They did not investigate H3K9Me3 at all. In addition, they did observe basal H3K27me3, but concluded that it is independent of Activin/Nodal signalling.

*Reviewer #2:*

*In this paper, the authors use several genome-wide approaches to investigate the mechanisms employed by Smad transcription factors to control transient gene expression profiles. A lot of data is generated and presented in this paper and from this the authors attempt to make several conclusions. However, overall, despite the large amount of work, the study lacks fundamentally new findings on how Smad transcription factors work. For example, the authors claim that Smad2 works through affecting polymerase recruitment and not pause release but the data are not entirely consistent with this view (see below). The authors also indicate that Smad binding kinetics do not necessarily correlate with gene expression profiles (and hence use different mechanisms at different genes). However, the authors do not take into account message stability which is potentially reshaping the target gene expression profiles rather than a transcriptional effect per se. Another important issue is the inclusion of "media only" effects in timecourses of activin signalling and this is not really appropriate (see below) and impacts on the interpretation of the gene expression profiles. Two of the latter findings (i.e. involvement of FOXH1 and SMARCCA4) are interesting but are already well known and studied Smad effector proteins. Overall, the current paper is therefore not conclusive enough for publication, in part through the inclusion of too much information that is not always necessary. However, with streamlining and careful re-interpretation of the data, a more coherent story might emerge.*

We thank the reviewer for their extremely useful insights. We have addressed all of the criticisms and comments with a large number of new experiments, as outlined in the point-by-point rebuttal below. We have also rewritten the manuscript to better communicate the novelty of the major conclusions.

The point of the study was to answer fundamental questions about how transcription factor binding regulates transcription. We wanted to discover whether transcription factor occupancy induces histone modifications, or if the chromatin landscape dictates recruitment. We wanted to define the sequence of events that occur from transcription factor binding to RNA Polymerase II regulation, and discover how transcription factor binding influences the temporal patterns of gene expression in the longer term. The best way to study these critical issues is to use a system where the transcription factor binds in response to extracellular signalling, and to do this we chose to focus on Activin/NODAL signalling.

In this paper for the first time we have delineated the sequence of events that occur from SMAD2 binding to transcriptional activation, and the mechanisms underlying them. We have now established three new paradigms for signal-dependent transcriptional regulation.

1) We have provided unequivocal evidence that transcription factor binding causes histone modification and chromatin remodelling with the discovery that SMAD2 binds to closed chromatin and remodels it.

This discovery also overturns the dogma in the field that SMADs act passively and only bind to ‘pre-conditioned’ chromatin.

2) We demonstrate that SMAD2 regulates RNA Polymerase II via de novorecruitment to target promoters. (See below for details about how we have corroborated these data.)

3) We demonstrate that SMAD2 chromatin binding does not directly correlate with transcription of target genes, and as a result have overturned the assumption in the field that transcription factor binding linearly equates with transcriptional kinetics.

We have now explored this latter issue in much more detail with an SB-431542 chase experiment (see Figure 1 and Figure 1—figure supplement 4). The results indicate that modulation of long-term responses to NODAL/Activin signalling requires on-going signalling. This provides a functional explanation for our observation that SMAD2 remains bound at regulatory regions of genes as their transcriptional profiles are modulated over time. It also supplies the previously missing crucial evidence that the underlying mechanism is via SMAD2 sequentially recruiting multiple regulators to chromatin.

*1) The inclusion of the "untreated" samples in parts B and C is not entirely valid. This is not the same as treating with activin for 48 hrs and is actually a different question being asked i.e. what is the difference between acute activin signalling responses and endogenously generated sustained signalling. This of course impacts greatly on the profiles in part C, where the last time point should be deleted. The "sustained" signal is not actually central to the story and could easily be deleted from the paper. Indeed, many of the later figures do not use this condition anyway.*

The reviewer raises a valid point. We wanted to include the chronic signalling condition for exactly the reason that the reviewer states, i.e. to understand the difference between acute signalling and endogenously-generated chronic signalling. To develop a comprehensive picture of how Activin/NODAL signalling through SMAD2 regulates transcription in both the short and long term, we think that this sustained signalling condition is essential. The most important insight it has given us is that SMAD2 remains bound to chromatin as long as signalling is active and directly contributes to the transcriptional responses (see also point 6below).

The issue we faced was how to plot the responses to chronic signalling on the same graphs as the acute signalling responses. We considered that it was valid to plot it as a 48 hr Activin time point as we found that the levels of pSMAD2 were the same, as were the target gene responses we measured. However, on further analysis (see Figure 10) we found that for other genes, 48 hr of Activin after an acute response is not the same as the ‘untreated’ state. Therefore, we have now just plotted the untreated sample by introducing a gap in the graphs (see new Figure 1, Figure 2, Figure 4, Figure 1—figure supplement 1 and Figure 2—figure supplement 2). That way we can directly compare the responses to acute and chronic signalling without implying that the untreated state is equivalent to a 48 hr Activin time point.

Author response image 1.24 hr and 48 hr of Activin treatment partially recapitulate the untreated condition for pathway activation and induction of target genes.(**A**) Western blot for pSMAD2, SMAD2/3 and TUBULIN (loading control) on lysates collected from cells treated overnight with SB-431542, followed by washout and stimulation for the indicated times with the concentrations of Activin shown, or left untreated from plating to harvest (48 hr). (**B**) qPCR on samples treated as in (**A**) for representative target genes. Plotted are the means and SEM of three independent experiments performed in duplicate. *** corresponds to a p-value of < 0.001; ** corresponds to a p-value of < 0.01; n.s., non-significant.**DOI:**
http://dx.doi.org/10.7554/eLife.22474.033

*2) Have the authors considered whether RNA stability is shaping the profiles in Figure 1? This could explain the difference between "induced sustained" versus "transient induced". If the authors look at the RNA polymerase profiles (as should report on transcription) across the same gene clusters, do they see the same effects?*

This is a very important point that we have now explored in more detail in the revised manuscript. To sum up our results, RNA stability obviously has an impact on the RNA-seq profiles, but is not responsible for the differences between ‘induced sustained’ and ‘transient induced’.

We have now measured the mRNA stability of a number of representative target genes and find no correlation between mRNA stability and the gene category (see new Figure 1—figure supplement 4 and subsection “Activin induces multiple temporal patterns of gene expression”, second paragraph). For example, *T* and *Tdgf1* are both very stable transcripts, yet *T* is a delayed gene and *Tdgf1* is in the ‘induced sustained’ category. Similarly, *Lefty1* and *Hes1* have relatively short half-lives, and are in the ‘induced sustained’ and ‘transient induced’ categories respectively.

Furthermore, we have also tested whether these different transcription profiles require on-going Activin signalling – an issue also relevant to point 6below. We find that the profiles of all of the genes tested, with the exception of *Id1* and *Id2,* require on-going signalling (see new Figure 1 and Figure 1—figure supplement 4). Particularly pertinent to the issue of mRNA stability, we find that all the ‘induced sustained’ genes we looked at require on-going signalling for their sustained expression. This proves that they are not simply in the ‘induced sustained’ category because their mRNA is stable.

Concerning the question of the RNA polymerase profiles, in the original version of the paper we had looked at the Pol II Ser5P profile across the gene clusters (Figure 4), and observed clear differences between the 4 categories. Most importantly, in the ‘transient induced’ category, the Pol II Ser5P enrichment for the majority of the genes decreases dramatically after 1 hr Activin as expected. For the ‘induced sustained’ category, for some of the genes the Pol II Ser5P enrichment remains high; for others there is a decrease in Pol II Ser5P enrichment after 1 hr (consistent with the observation that the rate of RNA production decreases after 1 hr for these genes – see subsection “Activin induces multiple temporal patterns of gene expression”, third paragraph), but it is not as pronounced as for the ‘transient induced’ genes.

*3) In many places, the authors include "representative experiments" rather than biological replicates. As "representative" implies replicates have been done, then why are they not included? Can the authors explain their rationale for doing this? For timecourse experiments, this can be acceptable due to adjacent timepoints acting as replicates (e.g. Figure 2) but for other experiments this is less clear. For example, Figure 1—figure supplement 2 should contain averages of the replicate experiments as this is a simple knockdown experiment. The same applies elsewhere and should be justified.*

All experiments have been performed at least twice and the vast majority three or more times. This is now explicitly stated in the Materials and methods (subsection “Statistical Analysis”).

In the revised version of the manuscript we have now presented combined experiments for the majority of experiments rather than representative ones (see new Figure 1, Figure 5, Figure 7, Figure 8, Figure 1—figure supplement 2, Figure 1—figure supplement 3, Figure 1—figure supplement 4, Figure 4—figure supplement 1, Figure 7—figure supplement 2, Figure 5—figure supplement 1, Figure 7—figure supplement 3 and Figure 8—figure supplement 1). For the data in Figure 5, Figure 7 and Figure 8, we show representative experiments to keep the spatial information and then plot the combination of the three experiments at the 5’ to SBS and 3’ to SBS (for the data in Figure 7 and Figure 8) or the SBS (for the data in Figure 5, Figure 7 and Figure 8) to be able to demonstrate the statistical significance of the data.

Note that for the spatial experiments like those shown in Figure 5, Figure 7, Figure 8 and Figure 8, if everything was normalized to the SB-431542 values, then the differences between the SB-431542 samples for different regions of the same gene and for different genes would be lost. It was to avoid these sort of losses of information resulting from normalization that we had originally opted to show representative experiments.

The only experiments that we are still showing as representative experiments are: time courses like Figure 2 for example, which as the reviewer states the adjacent timepoints act as replicates; and experiments which are validations of whole genome analyses, where we wanted to test the exact same mRNA or genomic DNA used for the NGS experiment.

*4) It is not clear how the replicate ChIP-seq experiments were processed to get the final list of Smad targets. How reproducible were the two ChIP-seq experiments? Also in generating "gene associated Smad activity" it is not clear why the authors took a "1bp" overlap in peaks. If looking at the same peak, then this is not stringent enough as the summits could be up to 500 bp apart and a parameter like summit-summit distance or% peak overlap should be used instead. If attempting to just associated total Smad binding with a gene, then the peak overlaps are not relevant. What happens if alternative ways of merging the Smad signal are used?*

Assuming that the reviewer is referring to the SMAD2 ChIP-seq experiments, then the replicates were very good, but the enrichment in the second biological replicate was significantly lower across the whole genome. This is obvious looking at the scales in Figure 2—figure supplement 1. Because of this, some of the smaller peaks were not called by MACS in the second dataset, but were present upon visual inspection. Thus, the final list of SMAD2 peaks was generated using the MACS-called peaks for the four time points of the first dataset. We have also validated a subset of them by qPCR (Figure 2—figure supplement 1). Note that when we defined the high confidence dataset of target genes and associated binding sites, we took into account the SMAD2 peaks conserved between the two biological ChIP-seq replicates within 100 kb of an annotated TSS or TTS (see subsection “Definition of the high confidence dataset of target genes and associated binding 877 sites”).

For the other ChIP-seq experiments, both biological replicates were used by the DiffReps program to identify the statistically significant loci of enrichment (see subsection “Pol II (Ser5P and Ser2P) and H3K27Ac/H3K9Ac ChIP-seq analysis”, first paragraph).

Concerning the merging of SMAD2 peaks. We used a minimum of 1 bp as an overlap, but as now stated in the text (see subsection “SMAD2 peak calling, annotation and generation of a SMAD2 consensus peak list”, second paragraph) the majority of peaks have a sizable overlap and the mean is 200 bp (see Figure 11). The idea behind this was not to define specific SMAD2 binding loci, but to define activity loci that applied across the different time points.

Author response image 2.**DOI:**
http://dx.doi.org/10.7554/eLife.22474.034

*5) Figure 3 is central to the authors' conclusions about whether pause-release is occurring or not. However, in the "baseline on" set, there is clear evidence of paused polymerase in the inhibited condition i.e. at the TSS. In contrast this is not the case in the "baseline off" samples. In the latter case, then "recruitment" must be the primary mechanism and in the former case, pause-release must be the initial event. After the initial events it is highly likely that both mechanisms are important in both cases (i.e. the polymerase must be recruited and move to the elongation phase at both types of site once the first polymerase has moved off). It is also apparent in Figure 3 that both Ser2 and Ser5 forms of the polymerase increase which again suggests both mechanisms are generally activated. Have the authors looked at total RNA polymerase levels by ChIP rather than the CTD marks which inevitably pick out the promoter-proximal and elongating polymerases.*

The reviewer says that there is clear evidence of a paused polymerase in the inhibited condition at the TSS of the ‘baseline on’ genes. I think there may be a difference in interpretation here. The reviewer appears to suggest that a peak of Pol II Ser5P at the TSS per seis a sign of a paused polymerase. However, this is not the case, and to clarify the situation we have now added metaprofiles for Pol II Ser2P (see revised Figure 3).

The profile expected for a paused polymerase is a peak of Pol II Ser5P at the TSS, with no enrichment in the gene body and also no Pol II Ser2P enrichment at all (Sawicka et al., 2014, PMID 25135956). For the ‘baseline off’ genes, the critical result is that there is no peak of Pol II Ser5P at the TSS and no Pol II Ser2P in the SB- 431542-treated state. Therefore, activation of these genes must be via Pol II recruitment, as the reviewer states (see new Figure 3).

The ‘baseline on’ genes, in contrast, in all conditions show the typical active gene profile (see new Figure 3), which is exemplified by the Pol II Ser5P and Pol II Ser2P profiles shown in Figure 3. These genes show a peak of Pol II Ser 5P at the TSS and TTS with enrichment in the gene body and enrichment of Pol II Ser2P in the gene body with a peak at the TTS. This is not consistent with a paused polymerase in the SB-431542-treated state, or in any other condition.

Most importantly, for these ‘baseline on’ genes, Activin/NODAL signalling does not change the shape of the Pol II Ser5P and Pol II Ser2P profiles. It simply changes the amplitude, which reflects changes in Pol II recruitment. This is also the case for genes in the other categories. See subsection “Activin/NODAL signaling regulates Pol II via recruitment”, second paragraph.

Thus, we conclude that the step regulated by NODAL/Activin signalling is Pol II recruitment.

We have not performed ChIP-seq with an antibody recognizing total RNA polymerase as we consider that using the antibodies specific to the different forms is more informative.

*6) Another issue arises from Figure 3 as it is clear that the "repressed" genes are actually transiently activated which is actually apparent from the RNAseq data. Interestingly these genes show a lot of evidence for paused polymerase at the TSS. Perhaps more importantly, looking back on Figure 1, all of the genes show increased activation at the 1 hr time point. This suggests a unifying mechanism of Smad action through upregulating the initial transcription levels. The authors invoke a mechanism of Smads recruiting additional co-repressors to then shape subsequent responses and there is no evidence presented to support this claim (and evidence should be provided for this claim to be made).*

We thank the reviewer for these comments as they inspired us to perform an SB- 431542 chase experiment that has been incredibly informative, and has provided critical evidence to support our claim that SMAD2 recruits additional factors to shape subsequent responses.

To discover whether Activin/NODAL signalling through SMAD2 induces an initial transcriptional up-regulation that is then modulated through other factors, or whether the long-term responses require continuous signalling, we treated cells for 1 hr with Activin and then with or without the type I receptor inhibitor SB-431542 for different times. The results are very striking (see new Figure 1 and new Figure 1—figure supplement 4 and subsection “Activin induces multiple temporal patterns of gene expression”, fourth paragraph).

Taking representative examples of target genes in each of the four categories we found that inhibiting signalling after 1 hr of Activin stimulation had a dramatic effect on the subsequent transcriptional profile of the target genes. ‘Induced sustained’ genes require continuous SMAD2 signaling for their long-term transcription, and ‘delayed’ genes require extended signaling to be induced at all. For ‘transiently induced’ genes like *Smad7* and *Hes1* we found that when signaling was terminated after 1 hr their levels fell rapidly to baseline, presumably because of the very short half-life of these mRNAs, whereas in the context of on-going signaling, these genes were repressed in a much more gradual fashion. Transcriptional repression of genes like *Fgfr2* also requires on-going signaling, although not for *Id1* and *Id2*.

These new data provide a functional explanation for the presence of continued SMAD2 on the regulatory regions of all these classes of genes. Crucially, these data demonstrate that for most genes, long-term modulation of the response requires SMAD2 activity and thus is likely elicited by SMAD2-mediated recruitment of additional factors.

*7) The authors claim that Smad has two modes of binding. One to open chromatin and one to closed chromatin. FAIRE data is provided to support this claim in a locus-specific manner (did the authors look genome-wide and what was the result?). Also, the authors plot H3 density across Smad binding regions. However, in Figure 5 there is very little change in H3 occupancy around the centre of the Smad binding region. Even in the "baseline off samples" (<30 regions), then the chromatin is still open at the Smad binding regions. Therefore, it appears that there is a window of accessibility already present to which Smad2 binds, and subsequently widens. Thus, the idea that Smad2 is binding differently is not really supported by the data but is in fact accessing a "nucleosome depleted" DNA region in all cases.*

We have now performed additional experiments and analyses to convincingly show that SMAD2 does exhibit two distinct modes of binding.

The reviewer considers that the chromatin is already open in the ‘baseline off’ samples (which is 29 genes, corresponding to 99 regions). This is not the case, as it is clear that Activin induction results in a dramatic decrease in H3 occupancy at these SBSs and also a substantial induction of acetylation in the flanking nucleosomes. To corroborate this result we subdivided this category of SBSs further to focus on those whose acetylation is highly induced upon signal induction (see Figure 6). In this new metaprofile (Figure 5, right hand panels) the depletion of H3 at the SBS is even more pronounced (more than 2-fold), as is the ligand-induced flanking acetylation.

Furthermore, we have now analysed by ChIP-PCR the H3 enrichment ± Activin induction for the genes for which we show the FAIRE. It is clear that H3 levels for *Lefty1* and *Pmepa1* are uniformly high at the SBS and 5’ and 3’ flanking nucleosomes in the absence of signalling and that Activin induction results in nucleosome displacement at the SBS (see new Figure 5—figure supplement 1). The nucleosome depletion at the SBSs for these two genes is also evident in the IGV profiles (Figure 5 and Figure 5—figure supplement 2).

Taking all these results together, we believe that we have convincingly demonstrated that activated SMAD2 binds to closed unacetylated chromatin, where it induces nucleosome eviction and histone acetylation.

We have not performed FAIRE genome-wide.

*8) Related to point 7, the authors indicated that FOXH1 is not "pioneering" as binds at the same time as Smad2. However, if they are binding in the same complex, could FOXH1 still not be "pioneering in that context" i.e. the one that would access the closed chromatin?*

This is an interesting issue. The point we were making was that FOXH1 is not pre-bound and acting as a ‘landing pad’ for the activated SMAD complex. Given that both FoxH1 and the activated SMADs bind simultaneously and they are interdependent on each other for binding, it is impossible to say which component is responsible for accessing the closed chromatin. One of the other key results in the paper is that SMARCA4 is required for opening up of the chromatin at some of these loci and it is SMAD2 that interacts with SMARCA4. We now favour a model of dynamic assisted loading as hypothesised by Swinstead et al. 2016 (PMID 27633730), which we mention in the first paragraph of the subsection “The role of SMAD2-cooperating transcription factors”.

*9) In the text, it is stated that H3K9Ac and K27Ac show the same patterns. However, they are clearly different in Figure 5. i.e. H3K9ac is transiently induced at 1hr at all sites whereas H3K27ac is sustainably induced. The former correlates with initial Smad binding and the initial induction of gene expression and suggests could be the critical event controlled by Smad2. It is not clear why the authors have not commented more on this or developed this line of investigation.*

Our data show clearly that the induction of H3K27Ac and H3K9Ac after 1 hr Activin are similar and correlate with initial SMAD2 binding (see metaprofiles for all SMAD2 peaks in Figure 5). The reviewer is right that the longer-term patterns are different for the two marks, possibly reflecting activities of different deacetylases. This is a very interesting issue, but we feel that it is beyond the scope of the current manuscript.

*10) It was not clear whether the selected genes in Figure 7 and Figure 8 are in the "baseline off" category as they should be if testing chromatin accessibility issues.*

*Lefty1*, *Lefty2*, and *Pmepa1*, are in the ‘baseline off’ category; the others are in the ‘baseline on’ category and are there for comparison. This is now made clear in the text (see subsection “SMARCA4 is required for nucleosome eviction at a subset of SMAD2 binding sites”). The categories for a given gene are in [Supplementary-material SD4-data].

*11) It is essential that biological replicates are included in Figure 8. At the moment for example it is difficult to conclude much from Figure 8, as SMad2 enrichment goes down irrespective of the gene (i.e. not as stated in the text). Statistical significance (or not) would help clarify this. The effect of SMARCA4 is not clearcut. In part D for example, the FAIRE signal changes at Pitx2 and Smad7 show similar changes despite different baselines. The interpretation of this data could be that SMARCA4 is important for maintaining open chromatin whether inducible by Smad2 or not (i.e. important for all open chromatin).*

We have now combined the biological replicates for both Figure 7 and Figure 8 (see also response to point 3 above and see Figure 7—figure supplement 3 and Figure 8—figure supplement 1). It is clear in Figure 8 that the dependency on SMARCA4 for transcription fits with the dependency for SMAD2 binding. We have clarified this in the text (see subsection “SMARCA4 is required for nucleosome eviction at a subset of SMAD2 binding sites”).

We have now combined the FAIRE data to be clear about the statistical significance (see Figure 8 and Figure 8—figure supplement 1). The reviewer is right, the FAIRE signal changes similarly at *Pitx2* and *Smad7* in the ligand-induced state despite the different baselines. However, the reduction in accessibility in the absence of SMARCA4 seen for these two genes is clearly not sufficient to affect ligand-induced transcription, SMAD2 binding or local histone acetylation (see Figure 8). The reviewer is also correct to say that there are effects of depleting SMARCA4 on the baseline, and thus it may have a general role in chromatin opening. Nevertheless, on top of this, SMARCA4 is essential in the case of the three ‘baseline off’ genes, *Lefty1, Lefty2*, and *Pmepa1,* for ligand-induced SMAD2 binding, histone acetylation, chromatin accessibility and transcription. We have now clarified this in the text (see subsection “SMARCA4 is required for nucleosome eviction at a subset of SMAD2 binding 476 sites”).

*Reviewer #3:*

[…] 1) Activin/nodal signaling induces phosphorylation of Smad2 and Smad3 (as described in the third paragraph of the Introduction of this paper). However, the authors analyzed binding of Smad2 to chromatin but not that of Smad3. Is it appropriate to discuss activin/nodal signaling only referring Smad2? They should explain in the main text why they omitted arguments about Smad3. Other ChIP-seq papers on activin/nodal signaling analyzed both Smad2 and Smad3.

*Yoon et al. Genes Dev. 25, 1654-1661, 2011 (not cited)*

*Kim et al. Dev. Biol. 357, 492-504, 2011 (cited)*

*Chiu et al. Development 141, 4537-4547, 2014 (not cited)*

The reason that we concentrated on SMAD2, was that there are undetectable levels of SMAD3 in these cells. This is obvious in all the Western blots in the paper where we used an antibody that recognizes SMAD2 and SMAD3 equally (BD 610843) (see for example, Figure 1 and Figure 1—figure supplement 1). To make this point even more strongly we have prepared a figure (Figure 12) where we have Western blotted extract from P19 cells alongside extracts from two other mouse cell lines, C2C12s and EpH4s. It is clear that SMAD3 is undetectable in P19s, but readily detectable in the other two lines. The levels of SMAD2 are comparable in all three lines. We have now stated in the text that we have focused on SMAD2, as that is the predominant R-SMAD downstream of Activin/NODAL in these cells (see Introduction, last paragraph).

Author response image 3.A) P19 cells express low level of SMAD3.Western blot for SMAD2/3 and TUBULIN (loading control) on lysates collected from three different mouse cell lines, the myoblast line, C2C12, the mammary epithelial cell line, EpH4 and P19 cells.Note that P19 cells express similar level of SMAD2 compared to the others cell lines, but undetectable levels of SMAD3. B) Characterization of the in house FOXH1 antibody. Lysates were collected from wild type P19 cells transfected with either non-targeting (NT) or *Foxh1* siRNAs and from P19 cells stably expressing MYC-tagged FOXH1 (see Figure 7—figure supplement 2). Shown are Western blots for FOXH1, MYC and TUBULIN (loading). Note that a band at the predicted molecular weight of FOXH1 is detected in the P19 MYC-FOXH1 sample when incubating the blot with either anti FOXH1 in-house antibody (left panel) or anti MYC antibody (right panel). Note that the levels of endogenous FOXH1 are below the threshold of detection in Western blots for the anti FOXH1 in-house antibody.**DOI:**
http://dx.doi.org/10.7554/eLife.22474.035

*2) To conclude that Smad2 regulates target gene expression through* de novo *recruitment of pol II (subsection “SMAD2 regulates Pol II via recruitment”, last sentence), the authors should directly show that knockdown of Smad2 results in failure of pol II to be enriched around the TSSs of target genes. I am afraid that Smad3 may also be involved. Same to the description in the fourth paragraph of the subsection “SMAD2 induces changes in the chromatin landscape to regulate transcription”.*

As explained above, we find SMAD3 to be undetectable in P19 cells. We show that for the ‘baseline off’ genes that there is no Pol II enriched at the TSS’s of target genes when the pathway is switched off with SB-431542, but that Pol II Ser5P becomes enriched at the TSS and in the gene bodies upon Activin stimulation. Because SMAD2 is the major R-SMAD, we ascribed the activity to SMAD2. However, to make the result more general we have rewritten the text to state that Activin/NODAL signalling regulates target gene expression through de novo recruitment of Pol II (see subsection “Activin/NODAL signaling regulates Pol II via recruitment”, last paragraph).

*3) How did the authors validate specificity of the anti-FoxH1 antibody prepared by themselves?*

We validated the antibody in several different ways. It is not quite sensitive enough to see endogenous FOXH1 in a Western blot. However, as we show in Figure 12, it clearly recognizes Myc-FoxH1 in a stably overexpressing P19 cell line by Western blot. It immunoprecipitates very effectively and most importantly, we see similar results in FoxH1 ChIPs in the absence and presence of ligand stimulation when we use our anti-FoxH1 antibody on wild type P19 cells or the Myc antibody in the P19 cell line that expresses Myc-FoxH1 at approximately endogenous levels (compare Figure 7 with Figure 7—figure supplement 2).